# Glutamate dehydrogenase from *Pantoea ananatis*: A new bacterial enzyme with dual coenzyme specificity

**Maria S. Kharchenko, Victoria S. Skripnikova, Julia G. Rostova, Natalia P. Zakataeva** *

Ajinomoto-Genetika Research Institute, Moscow, Russia

* natalia_zakataeva@agri.ru

## Abstract

Glutamate dehydrogenase (GDH) catalyzes the reversible conversion of α-ketoglutarate (α-KG) to L-glutamate (Glu) and plays an important role in linking nitrogen and carbon metabolism. However, little is known about this enzyme in the biotechnologically important bacterium *Pantoea ananatis*. In this study, we report for the first time the enzymatic characteristics of the *P. ananatis* AJ13355 GDH, Gdh$_{Pa}$. Gdh$_{Pa}$, a 46.5 kDa protein of the GDH50s group, was expressed in *Escherichia coli* host, purified, and biochemically characterized. *In vitro* enzymatic activity assays revealed that Gdh$_{Pa}$ is capable of catalyzing both the reductive amination of α-KG and the oxidative deamination of Glu with dual coenzyme specificity (NAD(H)/NADP(H)), a rare occurrence in bacterial GDHs. The NADH and NADPH specificity constants ($k_{cat}/K_m$) during reductive amination were similar (approximately 6 × 10$^4$), whereas the $k_{cat}/K_m$ value for NAD$^+$ was higher than that for NADP$^+$ during oxidative deamination (9.96 × 10$^3$ vs. 1.18 × 10$^3$). The only gene encoding GDH in *P. ananatis* AJ13355 (*gdhA$_{Pa}$*) is located on the megaplasmid pEA320 and has a low level of expression when cells are grown under ammonium- and glucose-rich conditions, indicating that *P. ananatis* does not assimilate ammonium via GDH under these conditions. Our studies show that *gdhA$_{Pa}$* expression increases significantly when cells are grown (i) under low glucose concentrations; (ii) using Glu, L-proline (Pro), L-arginine (Arg), or L-histidine (His) as the sole source of carbon; or (iii) using Glu, Pro, Arg, or His as the sole source of both carbon and nitrogen. These findings indicate that the *P. ananatis* GDH catalyzes the oxidative deamination reaction *in vivo*. The results of this study provide new insights into the function of GDH in *P. ananatis* and serve as a basis for future studies on the molecular mechanisms underlying the regulation of GDH activity in this biotechnologically important bacterium.

**Data availability statement:** All relevant data are within the paper and its Supporting Information files.

**Funding:** The author(s) received no specific funding for this work.

**Competing interests:** The authors have declared that no competing interests exist.

## Introduction

Glutamate dehydrogenase (GDH) catalyzes either the reductive amination of α-ketoglutarate (α-KG) to L-glutamate (Glu) using NAD(P)H as a coenzyme (anabolic reaction) or the oxidative deamination of Glu to α-KG using NAD(P)$^+$ as a coenzyme (catabolic reaction) (Fig 1) [1]. Whether the reductive amination or oxidative deamination reaction is carried out depends on several factors, including the host organism, physiological state, ecological niche, and nutritional source used. Oxidative deamination links amino acid metabolism to the tricarboxylic acid (TCA) cycle, whereas reductive amination allows for the uptake and incorporation of ammonium into Glu as an amino group, thereby supplying nitrogen to several biosynthetic pathways. Due to their important roles in nitrogen and carbon metabolism, GDHs have been found in all living organisms.

GDHs belong to the ELFV family of amino acid dehydrogenases, which also includes leucine (Leu), phenylalanine (Phe), and valine (Val) dehydrogenases. Based on the coenzyme required for the enzymatic reaction, GDHs can be divided into three subclasses: NAD-specific (EC 1.4.1.2), NAD/NADP (dual)-specific (EC 1.4.1.3), and NADP-specific (EC 1.4.1.4) [1]. The majority of catabolic GDHs are believed to be NAD-dependent, whereas anabolic GDHs are largely NADPH-dependent [2]. While most GDHs identified in microorganisms, fungi, and plants are either NAD(H)- or NADP(H)-specific, GDHs from higher animals and Archaea, as well as a few enzymes from prokaryotes, have dual coenzyme specificity [2,3]. Notably, NAD-specific, NADP-specific, and dual-specificity GDHs share remarkable structural similarity, with their coenzyme specificity often determined by just a few critical residues that can be interconverted through minimal mutagenesis. This functional plasticity suggests that NAD or NADP specificity has emerged multiple times during GDH evolution, with different lineages employing distinct structural solutions to achieve similar coenzyme preferences [2].

GDHs can be divided into three groups according to the molecular weight (MW) of the monomer, GDH50s, GDH115s and GDH180s, which are approximately 50 kDa, 115 kDa and 180 kDa, respectively. Currently known NADP- and NAD/NADP-specific GDHs belong solely to the GDH50s group, whereas NAD-specific GDHs have representatives in all three groups. The majority of GDHs are homo-oligomeric enzymes that differ in the number of monomers they contain [4–6]. Some GDHs may have

**Fig 1. Reaction catalyzed by GDH.**

hetero-oligomeric or higher-order multimeric structures that aggregate under certain conditions, such as high enzyme concentrations and high ionic strength [7,8]. While bacterial GDHs typically exist as homohexamers, eukaryotic GDHs have diverged into two families with distinct oligomeric architectures: fungi possess tetrameric GDHs composed of 115 kDa subunits, whereas vertebrates retain a homohexameric structure with 50 kDa subunits [9]. In the course of evolution, GDHs became a highly regulated enzyme with dual coenzyme specificity in mammals [2]. Plant GDHs form homo- or hetero-hexamers composed of subunits of approximately 45–50 kDa [10]. In *Arabidopsis thaliana* α, β, or γ subunits associate in different ratios, resulting in NAD-specific GDH isozymes with different kinetic properties and regulatory mechanisms. Hexamers composed of α and β subunits are present in roots, stems, and leaves, whereas those composed of α, β, and γ subunits are found only in roots. Recent studies indicate that organ- and metabolic-dependent changes in the ratio of the three isozyme subunits arose as a result of an evolutionary adaptation, in which whole-genome duplication resulted in the splitting of one GDH isozyme from a common ancestor shared by angiosperms and liverworts into GDHs corresponding to the α- and β-subunits [9]. This evolution of the plant GDHs allows for the maintenance of cellular homeostasis, adaptation to changing environmental conditions, balancing glutamate pools and fueling the TCA cycle with α-KG when carbon becomes the limiting factor, rather than serving for direct ammonium assimilation [11,12]. In addition to the three NAD-dependent GDHs, *A. thaliana* and certain other higher plants possess an NADP-specific enzyme that is evolutionarily closely related to bacterial NADP-dependent GDHs. This enzyme is localized in chloroplasts, and its precise physiological role remains unclear [13].

Michaelis-Menten constants of various GDHs for ammonium in the reductive amination of α-KG are generally quite high, and this low-energy cost reaction is used by cells under ammonia excess and low energy supply [14,15]. When cells grow under low ammonium conditions, ammonium is incorporated into Glu mainly via an energy-consuming pathway involving ATP-dependent glutamine synthetase (GS; EC 6.3.1.2) and NADPH-depending glutamate synthase (GOGAT; EC 1.4.1.13) (Fig 2). GDHs together with GOGAT have been shown to control Glu homeostasis and play a significant role at the branch point of the carbon and nitrogen assimilation pathways [3].

Most bacterial GDHs belonging to the GDH50s group are homo-hexamers with structurally similar monomers that possess two domains: (i) the N-terminal domain I, which is involved in substrate-binding and hexamer formation, and (ii) the C-terminal domain II, which contains a modified Rossmann fold and is involved in coenzyme binding [16,17]. The substrate binding site is located in a deep groove at the junction of the two domains, and during the enzymatic reaction, these two domains are brought closer together, thus narrowing the cleft and facilitating catalysis [16].

*Pantoea ananatis* is a gram-negative bacterium that, like *Escherichia coli*, belongs to the *Enterobacteriaceae* family and the *Gammaproteobacteria* class. *P. ananatis* strains have been isolated from many ecological niches and plant hosts and have been characterized as epiphytes, endophytes, and unconventional phytopathogens [18]. Moreover, some isolated nonpathogenic *P. ananatis* strains show promise as potential aids in the microbial production of useful substances, such as amino acids (Glu, L-cysteine, and L-aspartate), isoprenoids, etc. [19,20]. However, little is yet known about the regulation and enzymatic activity of players involved in the *P. ananatis* nitrogen assimilation pathway.

In this study, recombinant *P. ananatis* AJ13355 GDH protein (Gdh$_{Pa}$) was expressed in *E. coli*, purified, and biochemically characterized. Measurement of *in vitro* enzymatic activity revealed that Gdh$_{Pa}$ is capable of catalyzing both the reductive amination of α-KG and the oxidative deamination of Glu, demonstrating dual coenzyme specificity. The only gene encoding GDH in *P. ananatis* AJ13355, *gdhA$_{Pa}$*, is located on the pEA320 megaplasmid and has a low level of expression when cells are grown in minimal media rich in both ammonium and glucose. Its expression increases significantly under (i) glucose starvation conditions, (ii) when Glu, L-proline (Pro), L-arginine (Arg), or L-histidine (His) are used as the only source of carbon, and (iii) when Glu, Pro, Arg, or His are used as the only source of both carbon and nitrogen. These conditions, under which natural GDH activity occurs, indicate that Gdh$_{Pa}$ catalyzes the oxidative deamination reaction *in vivo*. The results of this study provide new insights into the function of the *P. ananatis* GDH and create a basis for further study of the molecular mechanisms underlying the regulation of GDH activity in this biotechnologically important bacterium.

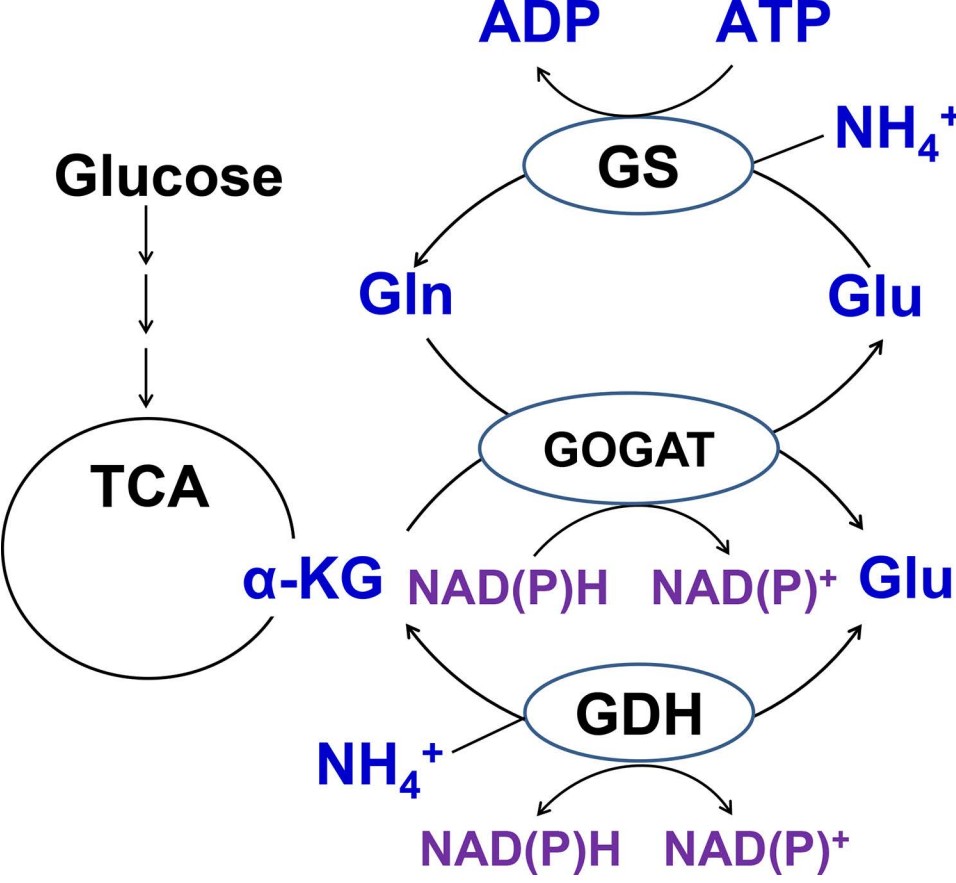

**Fig 2. Schematic representation of ammonium assimilation pathways in bacteria.**

## Materials and methods

### Bacterial strains, plasmids, and reagents

The bacterial strains and plasmids used in this study are shown in Table 1. *E. coli* strains TG1 and BL21(DE3) were used as hosts for cloning and protein expression, respectively.

All reagents were purchased from Sigma-Aldrich (Steinheim, Germany) unless otherwise specified.

### Culture conditions and preparation of crude cell extract

*E. coli* and *P. ananatis* were grown in Luria-Bertani (LB) or M9 minimal medium [21] supplemented with 1% D-glucose (unless otherwise specified). When required, ampicillin (Ap, 150 μg/mL) or tetracycline (Tet, 10 μg/mL) was added to the medium. Solid medium was obtained by adding 20 g/L agar to the corresponding liquid medium.

To examine the growth of *P. ananatis* cells using different carbon and nitrogen sources, cells grown on M9 agar medium with 1% D-glucose for 16 hours were collected, washed with 0.9% NaCl, and inoculated into 5 mL of either M9 or M9 without $NH_4Cl$ (M9-N) liquid medium supplemented with 1% D-glucose, 10 g/L Glu, 10 g/L Pro, 10 g/L Arg, or 10 g/L His, as specified, up to an initial optical density (OD) at 600 nm ($OD_{600}$) of 0.05. Cells were cultivated at 34°C with agitation at 70 rpm in a TVS062CA rocking incubator (Advantec, Tokyo, Japan). $OD_{660}$ was automatically monitored.

**Table 1. Bacterial strains and plasmids used in this study.**

| Strain or plasmid | Relevant characteristics[a] | Source or description |
|---|---|---|
| *E. coli* strains | | |
| TG1 | *supE hsdΔ5 thi Δ(lac-proAB)/F' traΔ36 proA⁺B⁺lacIq lacZΔM15* | VKPM[b] B5837 |
| BL21(DE3) | F⁻ *ompT hsdS_B* (r_B⁻, m_B⁻) *gal dcm* λ(DE3 [*lacI* P_lacUV5-T7*gene1ind1 sam7 nin5*] [*malB⁺*]_K-12 (λ^S) | Novagen |
| BL21(DE3)/pET-15-TEV-GdhA | BL21(DE3) containing expression plasmid | This work |
| *P. ananatis* strains | | |
| AJ13355 (SC(17)) | Wild type strain | [19] |
| SC17(0) | Derivative of SC17 resistant to the expression of λ Red genes | [22] |
| SC17(0)ΔgdhA::Tet | ΔgdhA_Pa derivative of SC17(0), Tet^R | Laboratory collection |
| Plasmids | | |
| pET-15b | *E. coli* expression vector, Ap^R | Novagen |
| pET-15–6His-GdhA(Pa) | pET-15b derivative with cloned *gdhA* gene from *P. ananatis* | This work |
| pET-15-TEV-GdhA(Pa) | pET-15b derivative for the production of Ht-TEV-GDH-Pan protein with an N-terminal hexahistidine tag and the tobacco etch virus (TEV) protease site | This work |

[a]Ap^R, ampicillin resistance; Tet^R, tetracycline resistance; and IPTG, β-D-l-thiogalactopyranoside

[b]VKPM, The Russian National Collection of Industrial Microorganisms

To determine GDH activity in crude cell extracts and to isolate RNA for reverse-transcription quantitative polymerase chain reaction (RT-qPCR) analysis, *P. ananatis* SC17(0) cells were grown at 34°C for 4 hours with aeration in M9 or M9-N liquid medium supplemented with D-glucose, Glu, Pro, Arg, His, $NH_4Cl$, $H_2O_2$, NaCl, and/or $(NH_4)_2SO_4$, or in medium E, pH 4.7 (10.2 g/L citric acid monohydrate, 26.7 g/L $K_2HPO_4$, 1 g/L $NH_4Cl$, 1 g/L NaCl, 1mM $MgSO_4$, 0.1mM $CaCl_2$, and 1% D-glucose), or in medium E, pH 7.0 (3.7 g/L citric acid monohydrate, 17.9 g/L $K_2HPO_4$, 1 g/L $NH_4Cl$, 1 g/L NaCl, 1 mM $MgSO_4$, 0.1 mM $CaCl_2$, and 1% D-glucose). All starter cultures were prepared by inoculating washed cells pre-grown for 16 h on M9 agar medium supplemented with 1% D-glucose to an initial OD600 of 4.

To prepare crude cell extracts, cells were first suspended in buffer A, consisting of 40mM Tris-HCl, pH 7.5; 1mM ethylene-diaminetetraacetic acid (EDTA); 10mM β-Mercaptoethanol; 10% glycerol; and 1mM 4-(2-aminoethyl)benzenesulfonyl fluoride hydrochloride (AEBSF). Cells were then lysed by sonication (2×40sec), and cell debris was removed by centrifugation at 13,200×*g* for 20min at 4°C. Protein concentration was estimated using a Bio-Rad DC Protein Assay kit (Bio-Rad, Hercules, CA, USA) according to the manufacturer's instructions.

## DNA manipulation

All recombinant DNA manipulations were conducted according to standard procedures [23] and the manufacturer's instructions. PCR amplifications and DNA sequence analyses were performed as previously described [24]. The oligonucleotide primers used in this study were synthesized by Evrogen (Moscow, Russia) and are listed in S1 Table. All constructs whose creation involved a PCR step were verified by DNA sequencing. Commercially available 1 kb DNA Ladder, Taq polymerase, T4 DNA ligase, High Fidelity PCR Enzyme Mix (Thermo Fisher Scientific Inc., Waltham, MA, USA), Gibson Assembly® Master Mix (New England Biolabs, Ipswich, MA, USA), and restriction enzymes (Thermo Fisher Scientific Inc., Waltham, MA, USA; New England Biolabs, Ipswich, MA, USA) were used. Plasmid DNA and DNA fragments were isolated using the Plasmid Miniprep and Cleanup Standard kit, respectively (Evrogen, Moscow, Russia). *E. coli* and *P. ananatis* cells were transformed with plasmids by electroporation using the MicroPulser Electroporator (BioRad Laboratories, Inc., Hercules, CA, USA).

## pET-15-TEV-GdhA(Pa) plasmid construction

A DNA fragment containing the native *P. ananatis gdhA* ($gdhA_{Pa}$; PAJ_RS21150) was obtained by PCR amplification using primers gdhAPA-NdeI-F and gdhAPA-BamHI-R and *P. ananatis* SC17(0) genomic DNA as a template. The resulting DNA fragment was digested with endonucleases *Nde*I and *Bam*HI and ligated to the digested pET-15b vector, yielding the pET-15–6His-GdhA(Pa) plasmid. The expression plasmid pET-15-TEV-GdhA(Pa) was obtained by replacing the thrombin cleavage site sequence in pET-15–6His-GdhA(Pa) with the TEV protease cleavage site sequence. This was done by generating a DNA fragment containing the $gdhA_{Pa}$ and pET-15b sequences by PCR amplification using pET-15–6His-GdhA(Pa) as a template and the primers 6His-TEV-R and gdhA_Pa-TEV-F, and then ligating it using the Gibson Assembly® Master Mix.

## Heterologous expression and purification of Gdh$_{Pa}$

*E. coli* BL21(DE3) cells were transformed with the expression plasmid pET-15-TEV-GdhA(Pa), yielding strain BL21(DE3)/pET-15-TEV-GdhA(Pa). The recombinant 6x-His-tagged *P. ananatis* GDH protein Ht-TEV-GdhA_Pan, which contains an additional 19 amino acid residues at the N-terminus (MGSSHHHHHHGT<u>ENLYFQG;</u> TEV protease cleavage site is underlined) was then produced in BL21(DE3)/pET-15-TEV-GdhA(Pa) cells via IPTG induction of the T7 promoter.

BL21(DE3)/pET-15-TEV-GdhA(Pa) cells grown on LB(Ap) agar plates with Ap were inoculated into 200 mL of LB(Ap) to an initial $OD_{600}$ of 0.15 and then cultured in a shaker to an $OD_{600}$ of 0.5–0.6. To induce protein expression, 1 mM of IPTG was added, and the cells were cultured at 34°C for 210 min. The cells were harvested by centrifugation at 6000 × *g* for 10 min at 4°C, washed with 0.9% NaCl, and resuspended in 10 mL of buffer B (20 mM Tris-HCl, 20 mM imidazole-HCl, and 0.5 M NaCl; pH 7.5). The cells were disrupted by sonication at 4°C until a clear lysate was obtained. Cell debris was removed by centrifugation at 13,200 × *g* for 20 min at 4°C. The Ht-TEV-GdhA_Pan protein was then purified by immobilized metal affinity chromatography (IMAC) on a HisTrap HP column (GE Healthcare, Uppsala, Sweden) as previously described [25]. Ht-TEV-GdhA_Pan-containing fractions were eluted with imidazole, combined, and loaded on a Sephadex G-25 column (Pharmacia) equilibrated with buffer containing 50 mM Tris-HCl at pH 7.5 and 25% glycerol. The protein was eluted in the same buffer, treated with TEV protease (Invitrogen™ AcTEV™ Protease, Thermo Fisher Scientific Inc., Waltham, MA, USA), and stored at −70°C until use.

Sodium dodecyl sulfate-polyacrylamide gel electrophoresis (SDS-PAGE) using 15% polyacrylamide gels was performed as previously described [26]. For native PAGE, a non-denaturing 4–20% gradient gel (Mini-PROTEAN® Precast Gels, Bio-Rad) was used. Protein bands were visualized using Coomassie Brilliant Blue R-250 staining. NAD$^+$-or NADP$^+$-dependent GDH activity was visualized using a stain consisting of 100 mM $KH_2PO_4$ buffer, pH 7.4; 150 mM Glu; 2 mM NAD$^+$ (or NADP$^+$); 1 mM nitroblue tetrazolium; and 0.5 mM phenazine methosulfate. The gels were incubated in this solution in the dark for 10 minutes at 37°C until dark blue bands appeared, washed with water, and fixed with 25% ethanol [27].

## Enzymatic assays

GDH activity was determined according to a previously described method [28]. Activity was measured in the oxidative deamination or reductive amination reactions at 25°C by monitoring increases or decreases in the absorbance at 340 nm ($\varepsilon = 6.22 \times 10^3$ M$^{-1}$ cm$^{-1}$), respectively. For the oxidative deamination activity assay, the reaction mixture (0.5 mL) contained 80 mM buffer (CHES, pH 9.5 for NAD$^+$ or imidazole, pH 6.0 for NADP$^+$), 150 mM Glu (unless otherwise specified), 2.0 mM NAD$^+$ (or NADP$^+$), and either purified recombinant Gdh$_{Pa}$ (1.11 µg) or crude cell extract (0.01–0.1 mg total protein). To determine their feasibility as potential substrates, we used 50 mM each of Pro, Arg, L-leucine (Leu), L-valine (Val), or L-phenylalanine (Phe) instead of Glu under these conditions. The nucleotides AMP, ADP, ATP, GMP, GDP, GTP, or IMP (1 mM each) and amino acids Pro, Arg, Leu, Val, or Phe (10 mM each) were individually tested as possible effectors. For the reductive amination activity assay, the reaction mixture (0.5 mL) contained 80 mM Tris-HCl buffer (pH 8.0), 5 mM α-KG, 50 mM ammonium chloride, 0.2 mM NADPH (or NADH), and either purified recombinant Gdh$_{Pa}$ (1.11 µg) or crude cell

extract (0.01–0.1 mg total protein). In each case, the reaction was initiated by the addition of a substrate. A control mixture without substrate was set up for each reaction. One enzyme activity unit (U) is defined as the amount of enzyme required to convert 1 μmol of substrate per min.

The pH dependence of GDH activity during both the oxidative deamination and reductive amination reactions was determined using purified recombinant $Gdh_{Pa}$. Assays were performed using 100 mM of the following buffers: MES buffer, pH 5.5–6.5; imidazole, pH 6.0–7.5; Tris-HCl, pH 7.1–8.9; and CHES, pH 8.6–10.0.

The kinetic parameters for purified recombinant $Gdh_{Pa}$ were determined using the appropriate activity assay with at least eight different substrate concentrations: 0–15 mM for α-KG, 0–0.5 mM for NADH/NADPH, 0–100 mM for $NH_4Cl$, 0–5 mM for NAD/NADP, and 0–250 mM for Glu. The data were analyzed using GraphPad Prism 10 software (GraphPad Software Inc., San Diego, CA, USA). The $k_{cat}$ values were calculated based on the subunit MW of recombinant $Gdh_{Pa}$. All kinetic parameters were obtained from at least three independent experiments.

## Mass spectrometry analysis

Mass spectrometry analysis of the protein samples was conducted in the Institute of Biomedical Chemistry, Moscow, Russia. Treatment of SDS-PAGE and PAGE gels, trypsin digestion, protein extraction, and mass analysis by time-of-flight matrix-assisted laser desorption-ionization (MALDI-TOF) were performed with a MALDI-TOF/TOF Bruker Ultraflex II mass spectrometer (Bruker, Germany) equipped with a delayed extraction system, reflection, YAG:Nd laser, and Bruker LIFT technology, as previously described [29]. Internal mass calibration was performed using the trypsin autolysis products. All spectra were obtained in reflector positive ion mode. The software package MASCOT (Matrix Science, Boston, MA, USA, http://www.matrixscience.com) was used to analyze the obtained peptide mass fingerprinting.

## RNA extraction, cDNA synthesis, and RT-qPCR analysis

Total RNA was extracted from cells grown as described earlier using ExtractRNA® reagent (BC032, Evrogen, Moscow) according to the manufacturer's instructions. To remove genomic DNA, the extracted RNA was treated twice with DNase I (Invitrogen™ Ambion™ DNase I, Thermo Fisher Scientific Inc., Waltham, MA, USA). The purity of the RNA was evaluated using a NanoDrop 2000 (Thermo Fisher Scientific, Sugarland, TX, USA) to determine the ratio of absorbance at 260 nm and 280 nm ($A_{260}/A_{280}$). To synthesize cDNA, reverse transcription (RT) was performed using the MMLV RT kit (SK021, Eurogen, Moscow) with 2 μg of total RNA and the reverse primers indicated in S1 Table. The synthesized cDNA was used as the template in the RT-qPCR amplification process, which was performed in a Rotor-Gene ™ 6000 thermocycler (Lab-gene, Archamps, France) using the ready-to-use mixture for PCR, 5X qPCRmix-HS (PK145S, Eurogen, Moscow, Russia), and the gene-specific primers for $gdhA_{Pa}$ and reference genes (listed in S1 Table). To test for contaminating DNA, an RT reaction in the absence of reverse transcriptase (no-RT control) was performed for each sample. The reference gene gyrB was used as an internal control to normalize the relative expression levels [30], which were calculated using the threshold ($2^{-\Delta\Delta Ct}$) method [31]. The reference gene representative of expression stability under the experimental conditions, gyrB, was selected using the Microsoft Excel application, NormFinder [32]. The best stability value obtained for gyrB was 0.261.

## *In silico* study and statistical analysis

A data bank search was conducted using the NCBI BLASTp algorithm [33]. A search for putative regulatory motifs in the upstream region of $gdhA_{Pa}$ was performed using SigmoID [34], BProm [35] Virtual Footprint [36], and EcoCyc database [37] tools. Data on experimentally characterized GDHs were extracted from the Brenda database (https://www.brenda-en-zymes.org/). Amino acid sequence alignments were performed using Clustal Omega and *ESPript* 3.0 programs [38,39]. Three-dimensional (3D) structures were obtained from the Protein Data Bank (PDB) or predicted with SWISS-MODEL (https://swissmodel.expasy.org/). The obtained 3D models were validated by the protein structure validation tool GMQE and QMEANDisCo Global on SWISS-MODEL. The models were further evaluated by SAVES v6.0 (https://saves.mbi.

ucla.edu/) and passed ERRAT, Verify3D, and PROCHECK scoring and testing. Structure alignments were performed in *PyMol* v3.1 (http://www.pymol.org/pymol). Phylogenetic analysis was performed using MEGA 12 software (http://www.megasoftware.net) [40].

All statistical analyses were performed using the GraphPad Prism 10 software as described.

## Results and discussion

### *In silico* search for genes involved in ammonium assimilation in *P. ananatis* AJ13355

*E. coli* has two alternative pathways for ammonia assimilation: (1) an NADPH-dependent GOGAT and ATP-dependent GS (encoded by *gltBD* and *glnA*, respectively) pathway, and (2) an NADPH-dependent GDH (encoded by *gdhA*) pathway [41]. Sequence analysis of the *P. ananatis* AJ13355 genome revealed an open reading frame (ORF), PAJ_RS17450, encoding a close homolog of *E. coli* GS, *glnA* (89% identity). While no ORFs encoding close homologs of *E. coli* GOGAT or GDH were found [19], a putative *gltB* gene (PAJ_RS19710) encoding a homolog with 64% identity to the large subunit of GOGAT from *Agrobacterium tumefaciens* (also known as *Rhizobium radiobacter*) was found to be located in the chromosome of AJ13355 [NCBI GenBank: AP012032.1]. In addition, a putative *gdhA* gene, *gdhA$_{Pa}$* (PAJ_RS21150), was found to be located in the pEA320 natural megaplasmid of AJ13355 [NCBI GenBank: AP012033.1].

Proteins greater than 90% identical to the predicted *gdhA$_{Pa}$* product, Gdh$_{Pa}$, have been found in many other *Pantoea* species, but their products have not yet been characterized. Among the characterized homologs, *E. coli* GDH (P00370) exhibits a relatively low amino acid sequence identity (27.3%), while GDHs from *Thermus thermophilus* (Q5SI04), *Thermotoga maritima* (P96110), and *Thermococcus profundus* [42–44] have higher identity (59%, 53.6%, and 46.6%, respectively; Fig 3). These homologous proteins have different properties and functions in their host cells. The *T. maritima* GDH demonstrated both reductive amination and oxidative deamination activities with dual NAD(H)/NADP(H) coenzyme specificity [43]. By contrast, the *T. thermophilus* GDH was characterized as an NAD-dependent Glu deaminating enzyme [42], and the *T. profundus* GDH was found to catalyze the NADPH-dependent synthesis of Glu [44].

Gdh$_{Pa}$ is annotated in the UniProt database (A0A0H3LAE4_PANAA) as a putative GDH. To preliminarily establish the coenzyme specificity of this enzyme, its primary sequence was compared to those of characterized GDHs (Fig 3). Certain consensus sequences in the cofactor binding domain of various dehydrogenases have been found to confer preferential coenzyme specificity, i.e., NAD(H) or NADP(H) [2,45]. The presence of an acidic residue (P7) near the glycine-rich motif GXGXXG (P1–P6) of domain II, a sequence "fingerprint" that determines coenzyme specificity in the widespread Rossmann fold of dehydrogenases [45,46] has been shown to discriminate against NADP. Dehydrogenases that exclusively use NADP(H) usually have the characteristic glycine-rich turn GSGXXA and a smaller, uncharged residue at P7 with positively charged residues nearby, allowing for better interaction with the 2′-phosphate of NADP(H). However, there are many exceptions to these "fingerprint" rules. NADP-specific *E. coli* GDH (EC 1.4.1.4) and other members of the GDH50s group with the same coenzyme specificity have Asp at P7 (Fig 3, S1A Fig), an amino acid which would not seem ideal in the adenosine phosphate-binding pocket. It was later shown that identical residues may confer different coenzyme specificity depending on the structural context [47]. Thus, the spatial arrangement and composition of not only the conserved amino acid residues at P1–P7 but also Ser at P8 and the positively charged residues Lys286/Arg289/Arg292 are responsible for the NADP-specificity of *E. coli* GDH. Unlike NADP-specific GDHs, which typically contain Ser or Ala at P2, NAD-dependent GDHs possess a large hydrophobic residue, usually Phe (S1A Fig), that sterically hinders the binding of the 2′-phosphate of NADP(H). Additionally, Ile (or other hydrophobic residues) at P8 further contributes to an environment incompatible with NADP(H) [11].

Phylogenetic analysis of 32 GDH50s-group proteins from diverse species identified six subfamilies (S1B Fig). Subfamilies IV and VI comprise NADP-specific enzymes from Eubacteria, plants, and Archaea, while the remaining four include NAD- or NAD/NADP-dependent GDHs found in plants (I), Eubacteria (II and III), and animals (including vertebrates) (V). The *E. coli* GDH clusters within the subfamily of NADP-specific bacterial and plant GDHs, whereas Gdh$_{Pa}$ groups with

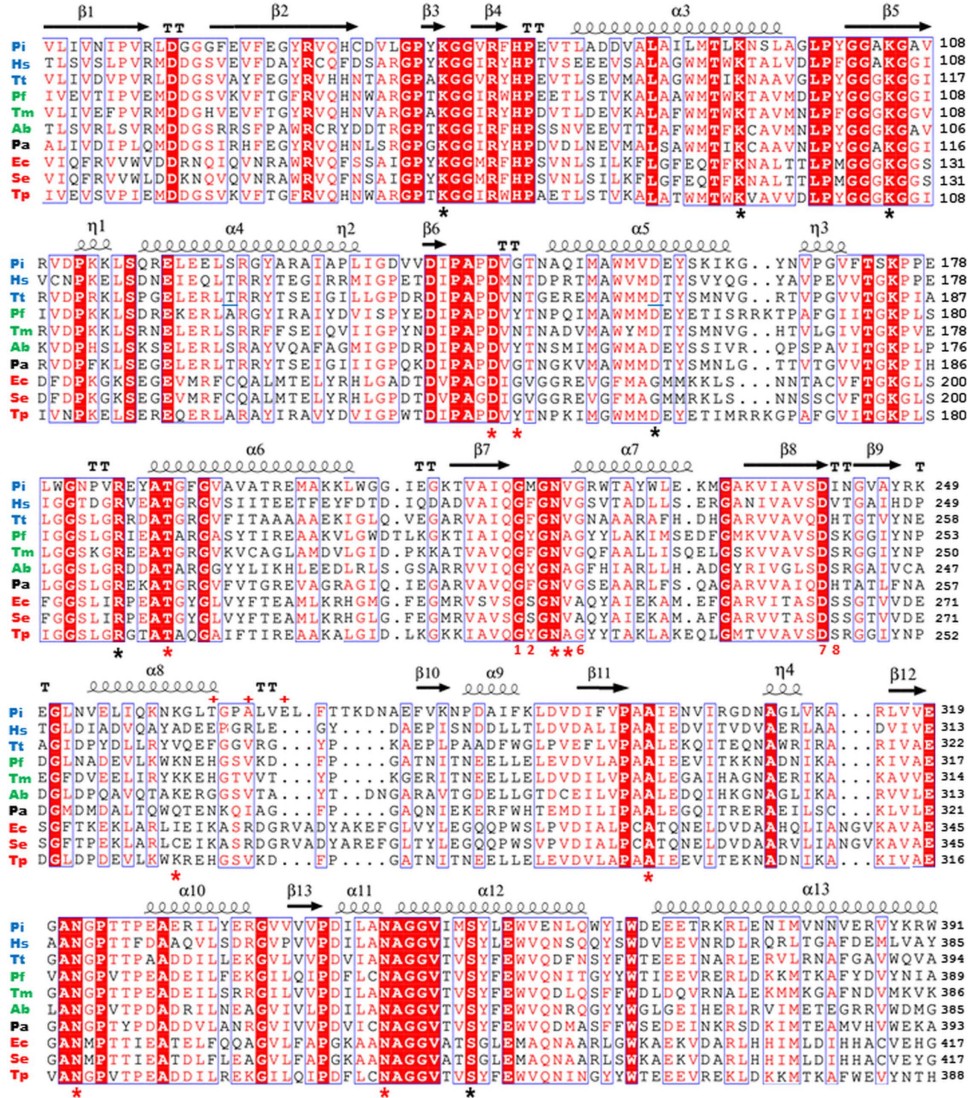

**Fig 3. Multiple sequence alignment of _P. ananatis_ GDH with several characterized bacterial GDHs.** Functionally significant sequence portions corresponding to _P. ananatis_ AJ13355 GDH (UniProt ID A0A0H3LAE4) (**Pa**); NAD(H)-dependent GDHs _Pyrobaculum islandicum_ (A1RVI3) (**Pi**), _Halobacterium salinarum_ (P29051) (**Hs**), and _Thermus thermophilus_ (Q5SI04) (**Tt**); NAD(H)/NADP(H)-dependent GDHs _Pyrococcus furiosus_ (P80319) (**Pf**), _Thermotoga maritima_ (P96110) (**Tm**), and _Azospirillum brasilense_ (A0A4D8RBL6) (**Ab**); and NADP(H)-dependent GDHs _Escherichia coli_ (P00370) (**Ec**), _Salmonella enterica_ serovar Typhimurium (P15111) (**Se**), and _Thermococcus profundus_ (O74024) (**Tp**). The secondary structure elements shown at the top of the alignment are based on the **Tp** structure (PDB ID 8XCS). Completely conserved amino acids are indicated in shaded red boxes. Numbers 1, 2, and 6–8 correspond to the conserved "fingerprint" amino acid residues P1, P2, and P6–P8, respectively, which are involved in coenzyme binding. Red asterisks (*) indicate additional residues that play an important role in coenzyme binding; black asterisks (*) indicate residues involved in catalysis and substrate recognition; red pluses (+) indicate positively charged conservative **Ec** residues K286/R289/R292; and underlining indicates residues in **Tt** responsible for Leu binding.

NAD- and NAD/NADP-specific GDHs and exhibits high sequence similarity to thermophilic bacterial homologs—a pattern consistent with horizontal gene transfer, particularly given the plasmid localization of the _gdhA_$_{Pa}$ gene. The phylogenetic clustering of Gdh$_{Pa}$, along with the presence of conserved NAD-dependent residues at positions P1–P6 (S1A Fig), suggests that Gdh$_{Pa}$ likely functions as an NAD- or NAD/NADP-dependent enzyme. However, shared ancestry and sequence

similarity alone are not definitive predictors of GDH coenzyme specificity [2], and so Gdh$_{Pa}$ was expressed, purified, and biochemically characterized to more confidently determine its coenzyme specificity.

### Heterologous expression, purification, biochemical characterization, and 3D structure modeling of recombinant *P. ananatis* GDH

To obtain the recombinant Gdh$_{Pa}$ protein, the 6x-His-tagged Gdh$_{Pa}$ protein, Ht-TEV-GDH-Pan, was expressed in *E. coli* BL21(DE3)/pET-15-TEV-GdhA(Pa) cells, purified by IMAC, and treated with TEV protease to remove affinity tag, which negatively affects enzymatic activity (S2 Table). The purity of the protein was then assessed using SDS-PAGE, which revealed a single protein band with a MW of approximately 45 kDa (S2 Fig). This correlates well with the predicted mono-mer MW of 46.6 kDa, indicating that homogeneous purification of Gdh$_{Pa}$ was achieved. Gradient non-denaturing PAGE followed by staining for GDH enzymatic activity with NAD$^+$ or NADP$^+$ as coenzymes detected protein bands correspond-ing to an apparent MW of approximately 260 kDa (Fig 4B, lanes 2 and 3, band #2). This is consistent with a hexameric structure for the active protein, which is typical of all GDHs of the GDH50s group. The origin of an additional, more diffuse band (Fig 4B, lanes 2 and 3, band #3) is unclear—it may be a pentameric form of the protein or hexamers with altered electrophoretic mobility. However, SDS-PAGE analysis of the same protein preparation showed a single band (Fig 4A, lane 1, band #1), and peptide mass fingerprinting analysis confirmed that bands #1, #2, and #3 exclusively contained the Gdh$_{Pa}$ protein with no detectable contaminating peptides above the noise threshold. This identification was highly signifi-cant, supported by Mascot scores of 344, 337, and 104 for bands #1, #2, and #3, respectively. The corresponding Expect values ($1.4 \times 10^{-29}$, $7.2 \times 10^{-29}$, and $1.4 \times 10^{-5}$) further validated these matches, indicating a low probability of random hits.

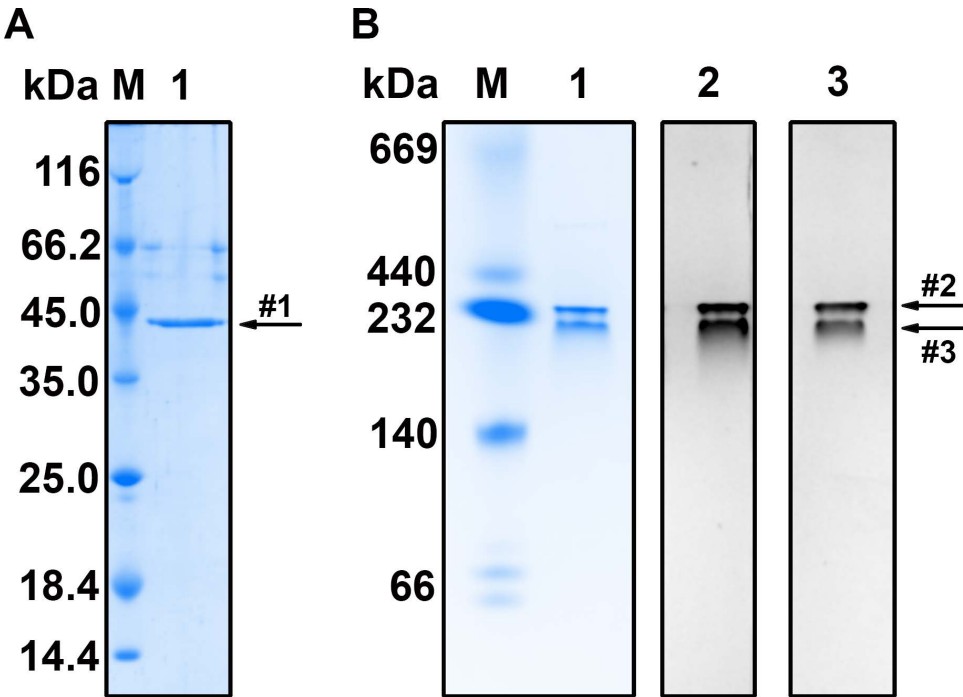

**Fig 4. SDS-PAGE and PAGE analysis of the purified recombinant Gdh$_{Pa}$.** (A) SDS-PAGE analysis. Lanes: M, marker proteins; 1, purified recom-binant Gdh$_{Pa}$ (1.6 µg). (B) Gradient non-denaturing PAGE. Lanes: M, native PAGE marker proteins; 1–3, purified recombinant Gdh$_{Pa}$ (1.6 µg) stained (1) with Coomassie Brilliant Blue R-250, (2) for NAD$^+$-dependent GDH activity, and (3) for NADP$^+$-dependent GDH activity. Bands #1, #2, and #3 were analyzed using MALDI-TOF mass spectrometry.

Additionally, the sequence coverage was 59%, 47%, and 23% for bands #1, #2, and #3, respectively, confirming the presence of Gdh$_{Pa}$ in all samples.

To characterize the protein, its specific GDH activity, optimum pH, and main biochemical constants were determined. The effect of the pH on Gdh$_{Pa}$ activity during both the reductive amination of α-KG and the oxidative deamination of Glu was evaluated (S3 Fig). The enzyme was found to be active in all studied NAD(H)- and NADP(H)-dependent reactions. The optimum pH for the reductive amination reaction was 8.0 regardless of the coenzyme used. However, the enzyme exhibited a coenzyme-dependent pH profile in the oxidative deamination reaction, with an optimum pH of 9.5 using NAD$^+$ as a coenzyme and 6.0–6.5 using NADP$^+$ as a coenzyme. The respective optimum pH for each reaction was used in subsequent studies of Gdh$_{Pa}$ activity.

Because Gdh$_{Pa}$ belongs to the ELFV family, its deamination activity using Pro, Arg, Leu, Val, and Phe as substrates was also investigated. No activity was detected in the presence of these amino acid substrates (i.e., activity was below the detection limit of 2.6 nmol mg$^{-1}$ min$^{-1}$). This result is consistent with the presence of three conservative γ-carboxylate-binding residues—K77 (RGPG<u>K</u>), R193 (GSLG<u>R</u>), and S356 (GVTV<u>S</u>)—in the Gdh$_{Pa}$ sequence that distinguishes the Glu-specific dehydrogenases from other ELFV family dehydrogenases [48] (Fig 3).

The kinetic parameters, Michaelis-Menten constants ($K_m$), specificity constants ($k_{cat}/K_m$), and maximum reaction rate ($V_{max}$) of recombinant Gdh$_{Pa}$ for different substrates (α-KG, NH$_4$Cl, and Glu) and coenzymes (NADH, NADPH, NAD$^+$, and NADP$^+$) in the reductive amination and the oxidative deamination reactions were measured. Analysis of the kinetic behavior of Gdh$_{Pa}$ in reactions with each of the tested substrates and coenzymes revealed that it followed Michaelis-Menten kinetics. In the presence of excess ammonium salt, Gdh$_{Pa}$ catalyzes the conversion of α-KG to Glu with approximately the same cofactor requirement, as indicated by the similarity between the specificity constants for NADH- and NADPH-dependent activities (approximately $6 \times 10^4$; Table 2). The enzyme is also able to use either NAD$^+$ or NADP$^+$ in the reverse reaction, though with greater activity in the presence of NAD$^+$—the specificity constant for NAD$^+$ was higher than that for NADP$^+$ by almost an order of magnitude ($9.96 \times 10^3$ vs. $1.18 \times 10^3$).

These characteristics indicate that Gdh$_{Pa}$ has dual coenzyme specificity and catalyzes the reversible conversion of α-KG to Glu in an NAD(H)/NADP(H)-dependent manner. It should be noted that $K_m$ values for Glu were significantly higher than those for α-KG and NH$_4$Cl and that $K_m$ values for NADH/NADPH were lower than those for NAD$^+$/NADP$^+$, suggesting the enzyme may predominantly catalyze cellular Glu biosynthesis. Consistent with these results, the apparent $K_m$ value for Glu (99.1 mM) was found to be significantly higher than those for other catabolic GDHs (shown in Table 3 and listed

**Table 2. Kinetic parameters of recombinant Gdh$_{Pa}$.**

**Reductive amination**

| NADH-dependent | | | | | NADPH-dependent | | | | |
|---|---|---|---|---|---|---|---|---|---|
| Substrate | $K_m$ [a] (mM) | $V_{max}$ [a] (U mg$^{-1}$) | $k_{cat}$ (s$^{-1}$) | $k_{cat}/K_m$ (s$^{-1}$ M$^{-1}$) | Substrate | $K_m$ [a] (mM) | $V_{max}$ [a] (U mg$^{-1}$) | $k_{cat}$ (s$^{-1}$) | $k_{cat}/K_m$ (s$^{-1}$ M$^{-1}$) |
| NADH | 0.070±0.014 | 5.43±0.34 | 4.21 | 6.01*10$^4$ | NADPH | 0.078±0.017 | 6.29±0.46 | 4.87 | 6.25*10$^4$ |
| α-KG | 1.30±0.21 | 5.93±0.26 | 4.60 | 3.54*10$^3$ | α-KG | 0.98±0.24 | 4.16±0.33 | 3.22 | 3.29*10$^3$ |
| NH$_4$Cl | 5.87±0.20 | 5.01±0.04 | 3.88 | 6.62*10$^2$ | NH$_4$Cl | 12.88±3.23 | 6.05±0.48 | 4.69 | 3.64*10$^2$ |

**Oxidative deamination**

| NAD$^+$-dependent | | | | | NADP$^+$-dependent | | | | |
|---|---|---|---|---|---|---|---|---|---|
| Substrate | $K_m$ [a] (mM) | $V_{max}$ [a] (U mg$^{-1}$) | $k_{cat}$ (s$^{-1}$) | $k_{cat}/K_m$ (s$^{-1}$ M$^{-1}$) | Substrate | $K_m$ [a] (mM) | $V_{max}$ [a] (U mg$^{-1}$) | $k_{cat}$ (s$^{-1}$) | $k_{cat}/K_m$ (s$^{-1}$ M$^{-1}$) |
| NAD$^+$ | 0.90±0.06 | 11.57±0.30 | 8.97 | 9.96*10$^3$ | NADP$^+$ | 0.44±0.05 | 0.67±0.02 | 0.52 | 1.18*10$^3$ |
| L-Glu | 99.1±15.9 | 10.48±1.24 | 8.12 | 8.20*10$^1$ | L-Glu | 27.6±5.4 | 0.50±0.08 | 0.39 | 1.14*10$^1$ |

[a]The kinetic parameters were determined by at least three independent GDH activity assays. Data are means±SEM, n=3.

in the BRENDA database), the majority of which have $K_m$ values ranging from 0.000011 mM to 49 mM. Moreover, $Gdh_{Pa}$ $K_m$ values for coenzymes and substrates involved in reductive amination fall within the reported range of corresponding $K_m$ values for NADPH-depending enzymes (Table 3, BRENDA database). $Gdh_{Pa}$ $K_m$ values for NADPH, α-KG, and $NH_4Cl$ were 0.078, 0.98, and 12.9, respectively, compared with their reported ranges of 0.038–1.64 mM, 0.0008–5.6 mM, and 0.033–144 mM, respectively.

The fact that $Gdh_{Pa}$ exhibited maximum specific activity during $NAD^+$-dependent oxidative deamination suggests that it can catalyze the degradation of Glu *in vivo*—e.g., under increased concentrations of intracellular substrate. Although $Gdh_{Pa}$ can use either $NAD^+$ or $NADP^+$ in this reaction, greater activity was observed in the presence of $NAD^+$, which is typical for metabolically active GDHs [2].

Further analysis of $Gdh_{Pa}$ coenzyme preference was performed using 3D structural comparison (Fig 5). The 3D structure homology models of $Gdh_{Pa}$ ($Gdh_{Pa}$(NAD) and $Gdh_{Pa}$(NADPH)) were generated using the SWISS-MODEL web server and validated as described in Materials and methods. The crystal structures of $NAD^+$-dependent *P. islandicum* GDH ($GDH_{Pi}$) in complex with NAD (PDB code: 1V9L) and NADPH-dependent *T. profundus* GDH ($GDH_{Tp}$) in complex with NADPH (PDB code: 8XCS) were used as templates for the $Gdh_{Pa}$(NAD) and $Gdh_{Pa}$(NADPH) models, respectively.

Structural comparison of $Gdh_{Pa}$(NAD) and $GDH_{Pi}$ in complex with NAD (Fig 5A) indicates that the overall design of the NAD-binding pocket is very well conserved in $Gdh_{Pa}$(NAD)—the exception being a substitution of Ile242 in $GDH_{Pi}$ with His250 in $Gdh_{Pa}$(NAD). His250 occupies a similar spatial position, thus maintaining the overall geometry of the

**Table 3. Kinetic parameters of bacterial GDHs with different coenzyme specificities.**

| Bacteria | | Subunit MW (kDa) | $K_m$ (mM) | | | | | Ref. |
|---|---|---|---|---|---|---|---|---|
| | | | NAD(H) | NADP(H) | Glu | $NH_4^+$ | α-KG | |
| **NADP** | *Escherichia coli* | 50 | | 0.060 (NADPH) 0.018 ($NADP^+$) | 2.30 | 2.53 | 0.68 | [49] |
| | *Lactobacillus fermentum* | 50 | | 0.078 (NADPH) 0.044 ($NADP^+$) | 79 | 6.76 | 5.6 | [50] |
| | *Salmonella typhimurium* | 48.6 | | 0.019 (NADPH) | 50 | 0.29 | 4.0 | [51] |
| | *Thermococcus profundus* | 43 | 0.076 (NADH) | 0.035 (NADPH) 0.077 ($NADP^+$) | 6.8 | 22 | 0.87 | [44] |
| **NAD** | *Bacillus cereus* | 48 | 0.56 ($NAD^+$) | | 7.4 | 96 | | [52] |
| | *Clostridium symbiosum* | 49 | 0.011 (NADH) | | | 61.1 | 0.31 | [53,54] |
| | *Halobacterium salinarum* | 45 | 0.34 ($NAD^+$) | | 11.9 | | | [55] |
| | *Pyrobaculum islandicum* | 36 | 0.025 ($NAD^+$) 0.005 (NADH) | | 0.17 | 9.7 | 0.66 | [56] |
| | *Thermus thermophilus* | 48 | 0.27 ($NAD^+$) | | 49 | | | [42] |
| **NAD/ NADP** | *Azospirillum brasilense* | 48 | 0.5 (NADH) | 0.013 (NADPH) 0.004 ($NADP^+$) | 10 ($NADP^+$) | 0.38; 100 (NADPH) 66 (NADH) | 0.25 (NADPH) 5 (NADH) | [57] |
| | *Pyrococcus furiosus* | 48 | 0.36 (NADH) | 0.012 (NADPH) | 0.60 | 6.0 | 0.33 | [58] |
| | *Thermotoga maritima* | 47 | 0.022 (NADH) | 0.058 (NADPH) | | 108 | 1.95 | [43,59] |
| | *Pantoea ananatis* | 46.5 | 0.9 ($NAD^+$) 0.070(NADH) | 0.078 (NADPH) 0.44($NADP^+$) | 99.1($NAD^+$) 27.6 ($NADP^+$) | 5.9 (NADH) 12.9 (NADPH) | 1.30 (NADH) 0.98 (NADPH) | This work |

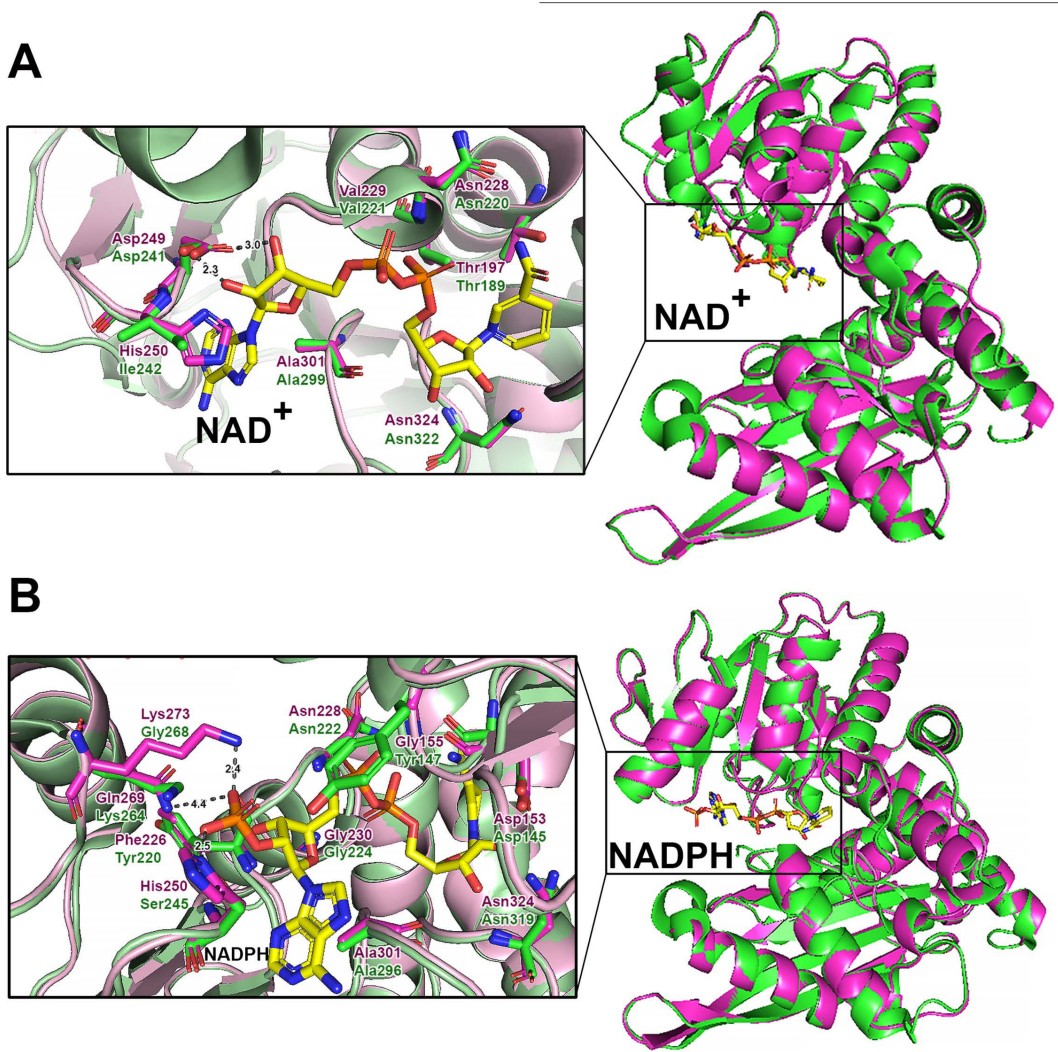

**Fig 5. Model of NAD and NADPH binding to Gdh$_{Pa}$.** (A) Structural model of Gdh$_{Pa}$(NAD), in magenta, superposed onto GDH$_{Pi}$, in green, complexed with NAD (PDB code: 1V9L). (B) Structural model of Gdh$_{Pa}$(NADPH), in magenta, superposed onto GDH$_{Tp}$, in green, complexed with NADPH (PDB code: 8XCS). In magnified views, residues interacting with the coenzyme (shown in stick representation) are labeled and indicated in green in the PDB structures, GDH$_{Pi}$ and GDH$_{Tp}$, while corresponding residues in the model structures, Gdh$_{Pa}$(NAD) and Gdh$_{Pa}$(NADPH), are indicated in magenta. The NAD and NADPH molecules bound to the active sites are shown in stick representation. The distances (Å) are represented by numbers and dashed lines. Figures were drawn using *PyMOL* v3.1.

NAD-binding pocket. While Ile is hydrophobic, His residues often participate in hydrogen bonding due to their imidazole side chains, which may further contribute to NAD-binding.

NADP differs structurally from NAD by the presence of an additional 2'-phosphate group on the adenyl ribose moiety. Analysis of the structural alignment of Gdh$_{Pa}$(NADPH) and GDH$_{Tp}$ in complex with NADPH suggests that Gdh$_{Pa}$ is likely capable of binding NADPH, despite the substitution of several residues known to stabilize the coenzyme in the active center of GDH$_{Tp}$ (Fig 5B). Structural modeling predicts several hydrogen bonds, and the two NADPH rings (adenine and nicotinamide) pack well against hydrophobic residues in the homology modeled GdhPa(NADPH) complex. Although the positively charged Lys264 that apparently enables electrostatic interactions with the additional 2'-phosphate group of

NADPH in GDH$_{Tp}$ is substituted by Gln269 in Gdh$_{Pa}$(NADPH), the proximal Lys273, located within 2.4 Å, is available for electrostatic interactions with the extra phosphate of NADPH. Ser P8 is strategically positioned to form a hydrogen bond with the 2'-phosphate, which is critical for NADP$^+$ recognition of NADPH-dependent GDHs and is thought to be a key inter-action missing in NAD$^+$-dependent enzymes [47]. While the P8 Ser of GDH$_{Tp}$ (Ser245) is substituted with His250 in Gdh$_{Pa}$, this substitution does not compromise NADP specificity, as the protonated imidazole ring of His maintains electrostatic interactions with the 2'-phosphate group under acidic conditions. However, under deprotonating conditions (i.e., basic pH), His may participate in hydrophobic interactions with the NAD ribose moiety and preferentially bind NAD over NADP (Fig 5A). Therefore, His250 may act as a pH-dependent "switch" for redox cofactor specificity, and this mechanism may under-lie the dual coenzyme specificity of Gdh$_{Pa}$ and explain the different pH optima for NAD$^+$- and NADP$^+$-dependent reactions (9.5 and 6.0–6.5, respectively) (S3 Fig) Similar observations have recently been reported for benzyl alcohol dehydroge-nase from *Aromatoleum aromaticum* [60].

Interestingly, although *T. thermophilus* GDH is described as strictly NAD-dependent [42], it has a His residue at P8 (Fig 3), indicating the potential for dual coenzyme specificity. Indeed, later studies have shown that this enzyme functions as a hetero-hexamer active in NAD(H)-dependent reactions during both reductive amination and oxidative deamination. Moreover, *T. thermophilus* GDH possesses NADP(H)-dependent activity, although at a rate three times lower than that of its NAD(H)-dependent activity [61]. It is therefore reasonable to speculate that His P8 facilitates the adaptation of GDHs to both redox cofactors, which may be useful in organisms that need to flexibly switch between NAD(H) and NADP(H) usage depending on metabolic conditions. The presence of His at P8 in GDHs may reflect evolutionary adaptation to enhance the enzyme's functional versatility in response to fluctuating metabolic demands or environmental stress.

Both *in vitro* and *in silico* characterization studies show that *P. ananatis* GDH has the potential to catalyze both the biosynthetic (aminating) and catabolic (deaminating) reactions *in vivo*, exhibiting dual coenzyme specificity in both cases. These properties are rarely found among the lower life forms, such as bacteria and yeasts, which generally possess GDHs with strict NAD(H) or NADP(H) specificity. It is thought that NAD-dependent GDHs catalyze mainly metabolic reac-tions, whereas NADP-specific enzymes are usually involved in biosynthetic pathways [2]. However, in higher eukaryotes, a single GDH possesses dual coenzyme specificity and can catalyze both anabolic and catabolic reactions under different physiological conditions [2].

The activity of animal GDHs is allosterically regulated by purine nucleoside phosphates and other metabolic intermediates. A key role in this allosteric regulation is played by the *antenna* helix, an α-helical nucleotide-binding domain near the C-terminus at the threefold axis in the GDH hexamer [62,63]. Contrary to those in higher eukary-otes, microbial GDHs do not have an *antenna* helix, and such levels of complexity in GDH activity have not been observed in microbes. However, a small number of microbial GDHs, particularly those from the GDH180s family, demonstrate allosteric regulation [5,6,64]. Arg and Asp were identified as positive allosteric effectors of the NAD$^+$-dependent GDHs from *Pseudomonas aeruginosa* and *Janthinobacterium lividum* [5,65]. AMP and ATP modulated activity of the NAD$^+$-depending GDHs from *Streptomyces clavuligerus* [6] and *Thiobacillus novellus* [66]. The Gdh$_{Pa}$ homolog *T. thermophilus* GDH, the active form of which is a hetero-complex of homologs GdhA and GdhB, is subject to allosteric activation by Leu [61].

To identify possible allosteric effectors of Gdh$_{Pa}$ activity, a number of amino acids and purine nucleotides were examined via oxidative deamination activity assay. However, none of the compounds screened had a noticeable effect on enzyme activity (S3 Table). While a slight increase in activity was observed for Leu (10 mM), this was only in the presence of unsaturated Glu (10 mM) (S3 Table). A sequence alignment of various GDHs revealed that the residues identified as potentially responsible for Leu binding in *T. thermophilus* GDH [61] are conserved in the Gdh$_{Pa}$ sequence (Fig 3). Additional experiments are required to study the role of Leu and other compounds as potential Gdh$_{Pa}$ effectors.

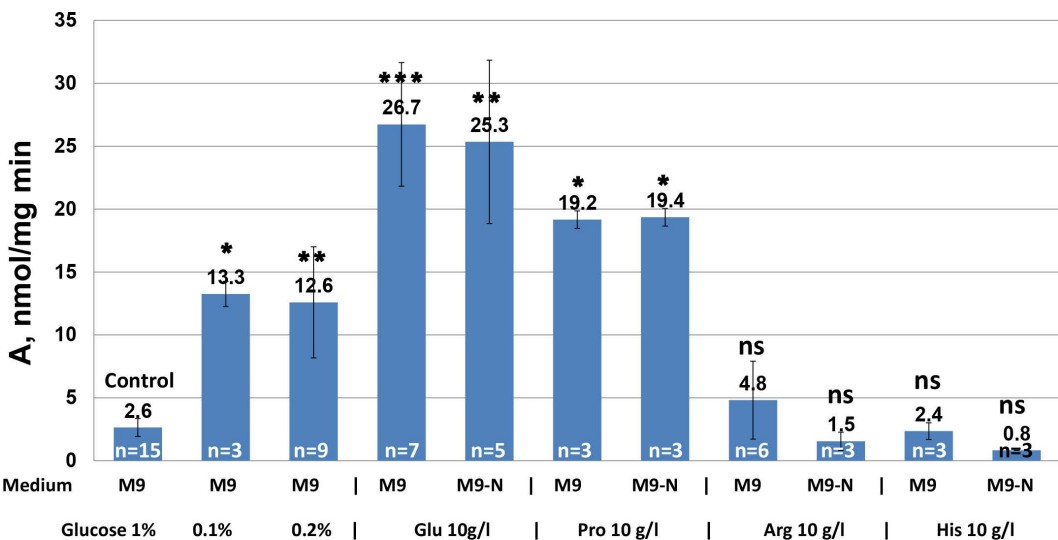

**Fig 6. GDH activity in *P. ananatis* cells under different growth conditions.** Specific GDH activities (per mg total protein) during NADPH-dependent reductive amination of α-KG in crude extracts of cells grown under indicated conditions. Data are means ± SD of several independent experiments (n); n is indicated on the corresponding bar. Brown-Forsythe and Welch ANOVA followed by Dunnett's post-hoc tests were used to determine statistically significant differences compared with the control group (Control); * $p < 0.05$, ** $p < 0.01$, *** $p < 0.001$, **** $p < 0.0001$, and n.s., not significant. Glu, L-glutamate; Pro, L-proline; Arg, L-arginine; and His, L-histidine.

When grown in minimal medium rich in both glucose and ammonium, *P. ananatis* SC17(0) was found to have very low GDH activity, with values close to the detection limit (Fig 6). Moreover, knockout of the *gltB* gene, which encodes GOGAT, led to Glu auxotrophy in this strain [19]. These data indicate that, contrary to *E. coli*, the main pathway for ammonia assimilation in *P. ananatis* involves GOGAT and GS but not GDH, even under high concentrations of ammonia.

### Native mechanism of the *P. ananatis* GDH activity

To elucidate the native mechanisms underlying GDH activity and its physiological role in *P. ananatis*, regulation of *gdhA*<sub>Pa</sub> gene expression was evaluated. Optimal growth conditions for native *gdhA*<sub>Pa</sub> expression were established by cultivating SC17(0) cells in liquid minimal media supplemented with different sources of carbon and nitrogen under different conditions (e.g., glucose starvation, varied pH values, and osmotic and oxidative stress). GDH activity in the NADPH-dependent reductive amination reaction was measured in the crude cell extracts to determine *gdhA*<sub>Pa</sub> gene expression at the translational level. These experiments revealed that neither alkalization and acidification of the culture medium nor osmotic and oxidative stress factors affected GDH activity (S4 Fig). However, GDH activity increased 5- to 10-fold under glucose starvation conditions and when Glu or Pro was used as the sole source of carbon or both carbon and nitrogen (Fig 6). Although we observed higher GDH activity in cells grown using Arg or His as the sole source of carbon or both carbon and nitrogen than in cells grown under control conditions (M9 medium supplemented with 1% glucose), there was no statistically significant difference between these values (Fig 6).

To evaluate *gdhA*<sub>Pa</sub> expression at the transcriptional level, RT-qPCR analysis was performed using RNA isolated from SC17(0) cells grown under the same cultivation conditions. Expression of *gdhA*<sub>Pa</sub> in SC17(0) cells grown under control conditions (M9 medium supplemented with 1% glucose) was compared with that in cells grown (i) under low glucose concentrations (0.1% and 0.2%), (ii) using Glu, Pro, Arg, or His (10 g/L each) as the sole source of carbon, or (iii) using

Glu, Pro, Arg, or His (10 g/L each) as the sole source of both carbon and nitrogen (Fig 7). The data showed that *P. ananatis* cells grown under the experimental conditions (i, ii, and iii) had $gdhA_{Pa}$ expression levels 5- to 36-fold higher than cells grown under high ammonium and glucose conditions (Fig 7). This suggests that the most likely physiological role for GDH in *P. ananatis* is the deamination of Glu under limited glucose conditions.

Sequence analysis revealed that the metabolic pathways that facilitate Pro, Arg, and His degradation in *P. ananatis* AJ13355 are the same as those that do in *E. coli*, and that the final product of the catabolism of these amino acids is Glu. The conversion of Glu to α-KG via GDH metabolic activity is required to couple these catabolic pathways to central metabolism and provide carbon- and nitrogen-containing compounds that can be rapidly metabolized for biosynthesis. Thus, the growth conditions leading to the induction of $gdhA_{Pa}$ expression indirectly dictate the metabolic function of $Gdh_{Pa}$ in *P. ananatis* cells. The fact that different amino acids have different effects on the expression of $gdhA_{Pa}$ (Fig 7) can be explained by differences in the uptake efficiencies of these exogenous amino acids, as well as by differences in the molecular mechanisms regulating both the expression of the corresponding catabolic genes and the expression of $gdhA_{Pa}$. Characterization of the molecular mechanism underlying the expression of $gdhA_{Pa}$ and identification of the regulatory factors involved require further study.

It is known that the NAD$^+$-dependent catabolic GDH from *Bacillus subtilis*, RocG, is involved in Arg and Pro catabolism and that *rocG* expression is induced by Arg, ornithine, and, to a lesser extent, Pro [67]. Similarly, expression of the *P. aeruginosa* PAO1 gene encoding NAD$^+$-dependent GDH, *gdhB*, was induced by the presence of Arg or Glu as the sole carbon and nitrogen source during growth. GdhB (MW = 182.6 kDa) is not homologous to $Gdh_{Pa}$ and belongs to a different class of GDHs, the GDH180s [5]. Transcription of *P. aeruginosa gdhB* is initiated via an Arg-inducible promoter and controlled by Arg regulatory protein (ArgR), a member of the AraC/XyIS family of regulatory proteins without homologs in *P. ananatis*. The *argR* gene (PAJ_RS15465) encodes the putative ArgR protein and shares 87.2% identity with the *E. coli* transcriptional regulator ArgR (characterized as a protein with multiple and complicated functions [68]).

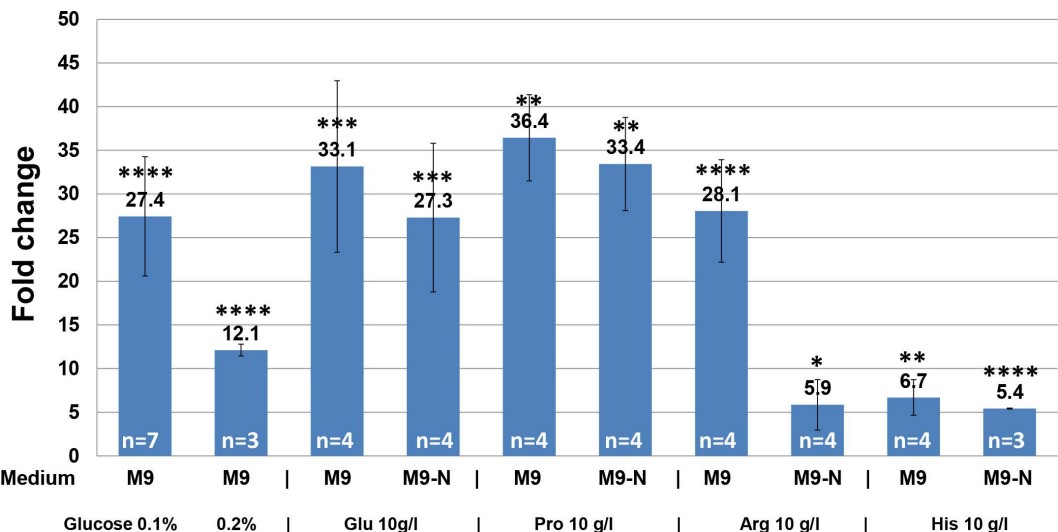

**Fig 7. Induction of *gdhAPa* expression under different growth conditions.** Relative $gdhA_{Pa}$ gene expression in cells grown under indicated conditions compared with cells grown under control conditions (M9 minimal medium supplemented with 1% glucose). Gene expression was determined using RT-qPCR and expressed as fold-change. Data are means ± SD of several independent experiments (n); n is indicated on the corresponding bar. Brown-Forsythe and Welch ANOVA followed by Dunnett's post-hoc tests were used to determine statistically significant differences; * $p < 0.05$, ** $p < 0.01$, *** $p < 0.001$, and **** $p < 0.0001$. Glu, L-glutamate; Pro, L-proline; Arg, L-arginine; and His, L-histidine.

Interestingly, this gene is present in the *P. ananatis* genome, and its protein product could be a possible regulator of $gdhA_{Pa}$ expression.

*In silico* analysis of the upstream region of $gdhA_{Pa}$ revealed sequences resembling those of binding sites for RNA polymerase with the alternative sigma factors sigma 32 (σ32) and sigma 24 (σ24) (S5 Fig). Moreover, sequences, which match well with the consensus −10 and −35 elements of the sigma 70 (σ70) promoter were revealed (S5 Fig). Considering the relatively low expression level in *P. ananatis*, this may indicate the involvement of some regulatory factors in $gdhA_{Pa}$ gene expression. Indeed, analysis using SigmoID, BProm, Virtual Footprint, EcoCyc database tools identified several putative binding sites for the regulatory proteins Nac, Crp, ArgR, NtrC, CpxR, and ArcA in the regulatory region of $gdhA_{Pa}$ (S5 Fig). Further experimental work is required to identify these factors and clarify their role in the regulation of $gdhA_{Pa}$ expression in *P. ananatis*.

## Conclusion

This study represents the first time that GDH was biochemically characterized in *P. ananatis*, a bacterium with bio-technological significance. The measured biochemical properties have revealed that the *P. ananatis* GDH is theo-retically capable of catalyzing both the reductive amination of α-KG using NADH and NADPH as coenzymes and the oxidative deamination of Glu using $NAD^+$ and $NADP^+$ as coenzymes, depending on metabolic conditions. This enzyme is encoded by a megaplasmid-localized gene, $gdhA_{Pa}$, and significant induction of gene expression was observed under conditions of glucose starvation, as well as when Glu or amino acids that are catabolized to Glu (i.e., Arg, His, and Pro) served as the sole source of carbon or both carbon and nitrogen. These data indicate that $Gdh_{Pa}$ catalyzes the deamination of Glu *in vivo*. Moreover, the disruption of $gdhA_{Pa}$ has no effect on, or even slightly improves, growth under ammonium- and glucose-rich conditions but impairs growth when Glu, Pro, and His are used as the sole sources of carbon and nitrogen (Fig 8).

However, the possibility that the *P. ananatis* GDH also catalyzes the anabolic reaction under certain conditions *in vivo* cannot be ruled out. The direction in which GDH carries out the reaction may depend on several factors, including sub-strate and coenzyme concentrations and enzymatic affinity ($K_m$ values) for these compounds. If $Gdh_{Pa}$ is involved in Glu catabolism as well as Glu biosynthesis and ammonium assimilation, a more complex mechanism to regulate its activity may be required to properly balance carbon and nitrogen metabolism. Additional factors, such as pH, the energy status of the cell, the development of specific cellular processes (for example, virulence) etc., can also determine the direction of the reaction. The allosteric regulation of GDH1, a mammalian GDH with dual coenzyme specificity, by both endogenous negative and positive modulators (i.e., GTP and ADP, respectively) serves as a versatile, energy-sensitive mechanism by which the enzyme can adapt to diverse and changing cellular environments [69]. The *antenna* structure of these enzymes found in higher eukaryotes also functions as an allosteric site. Despite lacking this *antenna* helix, some bacterial GDHs are also subject to allosteric regulation and require additional cofactors for their activity [5,6,49,64]. However, to our knowl-edge, homo-hexameric bacterial GDHs with dual cofactor specificity and allosteric regulation belonging to the GDH50s group have not yet been described. Elucidation of the molecular mechanisms underlying the regulation of *P. ananatis* GDH activity at both the transcriptional and post-translational levels requires further study. These mechanisms could pro-vide *P. ananatis* with flexibility in regulating metabolic changes and confer survival advantages under dynamic and stress-ful environmental conditions.

Protein engineering of dehydrogenase enzymes that can switch NAD/NADP coenzyme specificity is crucial for biotech-nology, as it allows for the optimization of metabolic pathways, enhances bioproduction efficiency, and expands the range of substrates that can be used in industrial and pharmaceutical applications [70–72]. Native dehydrogenases with dual NAD/NADP specificity play vital roles in cellular metabolism and enable organisms to maintain metabolic flexibility through dynamic regulation of redox balance and energy production, highlighting their evolutionary significance and potential as templates for future protein engineering efforts.

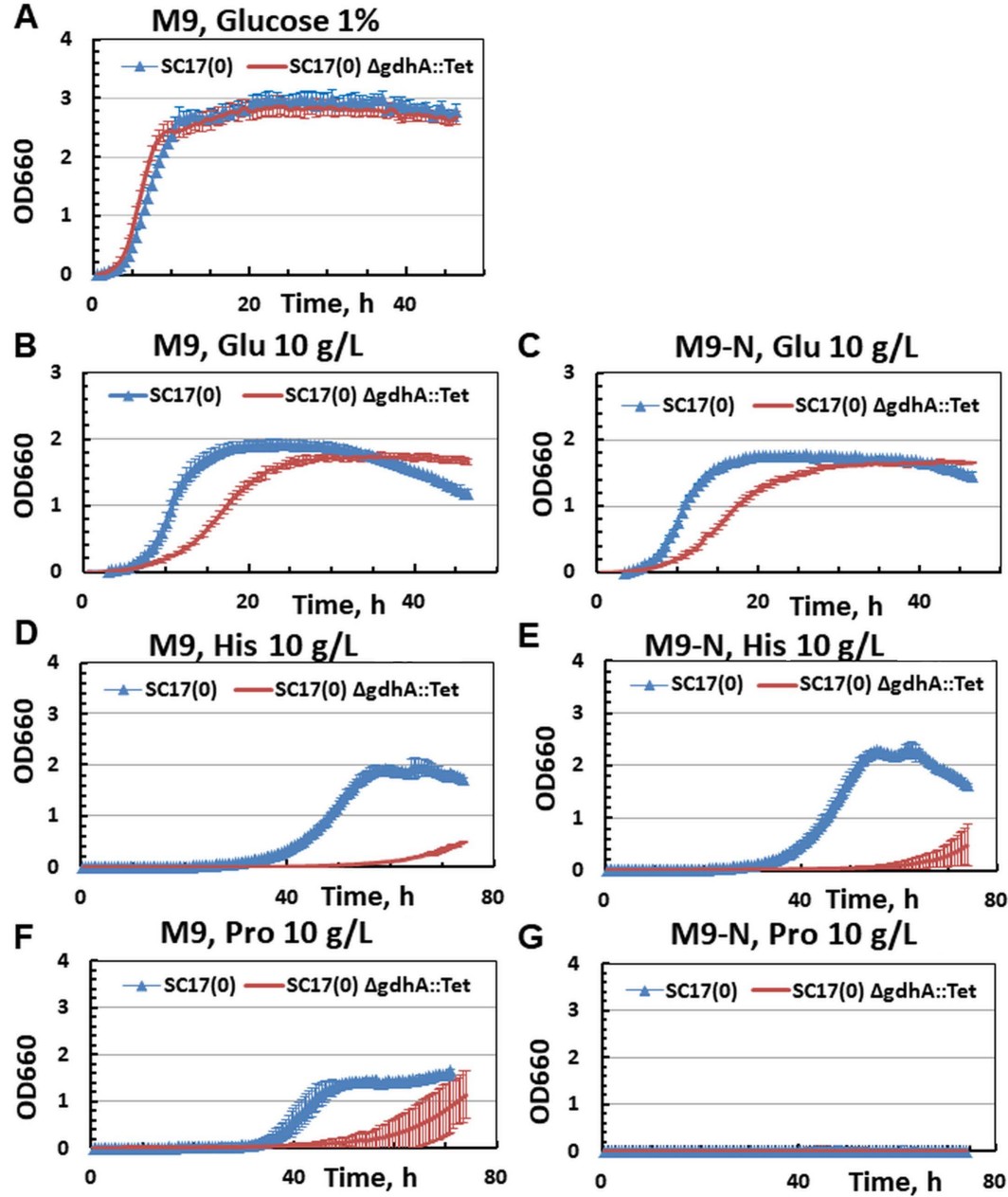

**Fig 8. Growth of SC17(0) and SC17(0)Δ*gdhA*::Tet in minimal medium.** (A) M9 with 1% glucose. (B) M9 with 10 g/L Glu. (C) M9-N with 10 g/L Glu. (D) M9 with 10 g/L His. (E) M9-N with 10 g/L His. (F) M9 with 10 g/L Pro. (G) M9-N with 10 g/L Pro. Data are means ± SD of three independent experiments. Some error bars are smaller than the data point symbols.

## Supporting information

**S1 Fig.** Phylogenetic analysis of GDH protein families. (A) Multiple sequence alignment of conserved coenzyme-binding motif of Gdh$_{Pa}$ and other species GDHs. The description of each protein sequence includes the protein name and species, and the UniProt or the GenBank accession number (in parentheses). NADP-, NAD- and NAD/NADP-dependent GDHs are

highlighted in pink, blue and green, respectively. Completely conserved amino acids are indicated in shaded red boxes. Numbers 1, 2, and 6–8 correspond to the conserved "fingerprint" amino acid residues P1, P2, and P6–P8, respectively. (B) Neighbor-joining tree for 32 GDH50s-group proteins from various species generated by MEGA 12 software. The bootstrap supports (%) calculated from 1,000 replicates are indicated next to the branches. Subfamilies are indicated by Roman numerals (I-VI).

(TIF)

**S2 Fig.** SDS-PAGE analysis of protein fractions of recombinant *P. ananatis* GDH. SDS-PAGE was performed on a 15% acrylamide gel, and the proteins were visualized using Coomassie Brilliant Blue R-250 staining. Lanes 1, 3: the purified Ht-TEV-GDH_Pan; lanes 2, 4, 5: aliquots of reaction of affinity tag removal from the purified Ht-TEV-GDH_Pan; lanes 6, 7: the purified Gdh$_{Pa}$ (2 µg and 1.6 µg, respectively); and m: marker proteins. MWs in kDa are shown on the right.

(TIF)

**S3 Fig. pH dependence of Gdh$_{Pa}$ activity.** Activity was measured in both NAD(P)H-dependent (reductive amination) and NAD(P)$^+$-dependent (oxidative deamination) reactions in various buffers (100 mM): MES, pH 5.5–6.5; imidazole, pH 6.0–7.5; Tris-HCl, pH 7.1–8.9; and CHES, pH 8.6–10.0. Data are means ± SD of three independent experiments. Some error bars are smaller than the data point symbols.

(TIF)

**S4 Fig. GDH activity in *P. ananatis* cells grown under different stress conditions.** Specific GDH activities (per milligram of total protein) during NADPH-dependent reductive amination of α-KG, measured in cells grown under the conditions indicated (different pH values, high ammonia supply, and addition of inhibitory concentrations of NaCl or $H_2O_2$). Data are means ± SD of several independent experiments, n = 3. Brown-Forsythe and Welch ANOVA followed by Dunnett's post-hoc test were used to determine statistically significant differences between the given group and the control group (Control); $p > 0.05$, not significant.

(TIF)

**S5 Fig. Sequence of the regulatory region of *gdhA$_{Pa}$*.** Putative regulatory elements were identified using various *in silico* tools. Putative −10 and −35 promoter sequence for σ70 (green font), putative promoter sequences σ32 (orange box), and σ24 (pink box) are indicated. Putative regulatory binding sites for CpxR (italics), Crp (red font), ArcA (underlined), ArgR (bold), NtrC (blue font), and Nac (blue boxes) are also marked. The *gdhA$_{Pa}$* coding sequence is indicated in red lowercase letters.

(TIF)

**S1 Table. Primers used in this study.**
(DOCX)

**S2 Table. Comparison of specific GDH activities of purified proteins before (Ht-TEV-GDH-Pan) and after (Gdh$_{Pa}$) affinity tag removal.**
(DOCX)

**S3 Table. Effects of amino acids and purine nucleotides on GDH activity.**
(DOCX)

## Acknowledgments

We sincerely thank Dr. Katashkina J.I. for donating the SC17(0)ΔgdhA::Tet strain.

## Author contributions

**Conceptualization:** Maria S. Kharchenko, Natalia P. Zakataeva.

**Data curation:** Maria S. Kharchenko, Victoria S. Skripnikova, Julia G. Rostova, Natalia P. Zakataeva.

**Formal analysis:** Maria S. Kharchenko, Victoria S. Skripnikova, Julia G. Rostova, Natalia P. Zakataeva.

**Investigation:** Maria S. Kharchenko, Victoria S. Skripnikova, Julia G. Rostova.

**Methodology:** Maria S. Kharchenko, Victoria S. Skripnikova.

**Project administration:** Natalia P. Zakataeva.

**Supervision:** Natalia P. Zakataeva.

**Validation:** Maria S. Kharchenko, Victoria S. Skripnikova, Julia G. Rostova, Natalia P. Zakataeva.

**Visualization:** Maria S. Kharchenko, Victoria S. Skripnikova, Julia G. Rostova, Natalia P. Zakataeva.

**Writing – original draft:** Maria S. Kharchenko, Victoria S. Skripnikova, Natalia P. Zakataeva.

**Writing – review & editing:** Maria S. Kharchenko, Julia G. Rostova, Natalia P. Zakataeva.

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
