## [Decision Letter · Decision Letter 0]

Dear Dr. Zakataeva,

Thank you for submitting your manuscript to PLOS ONE. After careful consideration, we feel that it has merit but does not fully meet PLOS ONE’s publication criteria as it currently stands. Therefore, we invite you to submit a revised version of the manuscript that addresses the points raised during the review process.

Authors are required to address all feedback provided by the reviewers, with particular emphasis on the need for revisions in the Introduction section. Additionally, any problematic aspects of the methodology must be rectified, along with the parameters selected for discussion. It is imperative to consider the statistical components, as this was frequently highlighted by the reviewers. Furthermore, considerations regarding the zymogram, which is crucial for the publication of the article, must also be duly acknowledged.

We look forward to receiving your revised manuscript.

Kind regards,

Marcos Pileggi, Ph.D

Academic Editor

PLOS ONE

Reviewers' comments:

Reviewer's Responses to Questions

**Comments to the Author**

1. Is the manuscript technically sound, and do the data support the conclusions?

Reviewer #1: Yes

Reviewer #2: Yes

2. Has the statistical analysis been performed appropriately and rigorously?

Reviewer #1: I Don't Know

Reviewer #2: No

3. Have the authors made all data underlying the findings in their manuscript fully available?

Reviewer #1: No

Reviewer #2: Yes

4. Is the manuscript presented in an intelligible fashion and written in standard English?

Reviewer #1: No

Reviewer #2: No

Reviewer #1: The study provides a novel characterization of a glutamate dehydrogenase (GDH) from Pantoea ananatis, a bacterium with biotechnological relevance. The dual coenzyme specificity of GdhPa (NADH/NADPH in reductive amination and NAD+/NADP+ in oxidative deamination) is a rare feature among bacterial GDHs.

Find my comments,

-The manuscript is mostly clear but contains minor grammatical errors and awkward phrasing.

-Introduction section should present importance of NAD(P) dependence for dehydrogenases. A comparison to other DHs in a table would strengthen this section.

Here some references for review;

"Cloning and expression heterologous alanine dehydrogenase genes: Investigation reductive amination potential of L-alanine dehydrogenases for green synthesis of alanine derivatives", Heliyon, 1(5), 2024, e26899. "Application of reductive amination by heterologously expressed Thermomicrobium roseum L-alanine dehydrogenase to synthesize L-alanine derivatives, Enzyme and Microbial Technology, 2023, 169, 110265. "NADP+ dependent formate dehydrogenase: A Review" Biocatalysis and Biotransformation, 2020, 39(4), 260-268.

-The selection of buffers (MES, imidazole, Tris-HCl, CHES) is appropriate, but the rationale for using pH 6.0 for NADP+ and pH 9.5 for NAD+ in oxidative deamination should be explicitly discussed.

-It would be helpful to compare kinetic parameters to a broader set of bacterial GDHs, emphasizing how GdhPa's properties differ functionally.

- The mechanistic insights into dual coenzyme specificity could be explored further, possibly through structural analysis or computational modeling. tructural modeling of GdhPa using Alphafold2 or SWISS-MODEL would provide deeper insights into coenzyme binding and active site interactions. Here a reference for reviewing, "Structural insights into the NAD+-dependent formate dehydrogenase mechanism revealed from the NADH complex and the formate NAD+ ternary complex of the Chaetomium thermophilum enzyme" Structural Biology, 2020, 212(3),107657.

-The statistical significance of differences should be clearly indicated (p-values, error bars clarification, etc.).

Reviewer #2: An English language revision is required.

The Introduction, as well as the literature, require improvement: mention of plant GDHs and their evolution must be included (see for reference: Kwinta, J., Bielawski, W. Glutamate dehydrogenase in higher plants. Acta Physiol Plant 20, 453–463 (1998). https://doi.org/10.1007/s11738-998-0033-1. Tercé-Laforgue et al. Resolving the role of plant glutamate dehydrogenase: II. Physiological characterization of plants overexpressing the two enzyme subunits individually or simultaneously. Plant Cell Physiol. 2013 Oct;54(10):1635-47. doi: 10.1093/pcp/pct108. Brambilla et al. Glutamate dehydrogenase in “Liverworld”—A study in selected species to explore a key enzyme of plant primary metabolism in Marchantiophyta. Physiologia Plantarum 2023, 175, doi: 10.1111/ppl.14071)

Results:

1) statistical analysis and significance must be performed -and provided - for the enzyme activity results

2) a more detailed discussion about the conserved aminoacids in functionally critical positions must be provided

3) in 2025, no newly characterized GDHs can lack an in gel activity characterization, as zymogram is the only way to reveal and assess the distribution of GDH isoenzymes in their active form. This is my major point, and it's not questionable....

Please fix "Mechaelis constants" with "Mechaelis-Menten constant"

Figure 4 and 5: please delete "additives"

Figure S1: please provide a less slovenly image

Figure S2: data are presented without standard deviation bars and statistically significance marks. Please provide a more scientific graph.

Figure S3: standard deviation bars curiously seem too homogenous among each others.... My feeling is that bars do not represent the real deviation from means, as the statistical analysis is apparently lacking. Please provide the relative test applied and the significance

Figure S5: in my opinion. it should be embedded in the main text

Supporting Table S1: primer's names are quite weird, as normally they are not marked with + o - symbols rather than with F o Fw (for forward) and R o Rev (for Reverse)

**Do you want your identity to be public for this peer review?** For information about this choice, including consent withdrawal, please see our Privacy Policy

Reviewer #1: No

Reviewer #2: No

---

## [Author Response · Author response to Decision Letter 1]

13 May 2025

Response to Reviewers

Reviewer #1:

The authors would like to express their gratitude to Reviewer #1 for the valuable comments and constructive suggestions provided for the revision of the manuscript. We have carefully addressed all remarks, and our detailed responses are presented below

Substantive revisions made in response to the reviewers' comments are highlighted in yellow in the file titled "Revised Manuscript_partially_marked". Additionally, all corrections—including both substantive changes and grammatical/stylistic edits—are tracked in the "Revised Manuscript with Track Changes" file, generated using Microsoft Word’s Compare Documents feature.

The line numbers mentioned in the response correspond to the documents "Manuscript” and "Revised Manuscript_partially_marked".

The study provides a novel characterization of a glutamate dehydrogenase (GDH) from Pantoea ananatis, a bacterium with biotechnological relevance. The dual coenzyme specificity of GdhPa (NADH/NADPH in reductive amination and NAD+/NADP+ in oxidative deamination) is a rare feature among bacterial GDHs.

Find my comments,

Comment: -The manuscript is mostly clear but contains minor grammatical errors and awkward phrasing.

Response: The text has been thoroughly revised for English language accuracy using editing service LetPub (http://www.letpub.com). A large number of grammatical and stylistic corrections have been made as indicated in the file “Revised Manuscript with Track Changes”.

Comment: -Introduction section should present importance of NAD(P) dependence for dehydrogenases. A comparison to other DHs in a table would strengthen this section.

Here some references for review;

"Cloning and expression heterologous alanine dehydrogenase genes: Investigation reductive amination potential of L-alanine dehydrogenases for green synthesis of alanine derivatives", Heliyon, 1(5), 2024, e26899. "Application of reductive amination by heterologously expressed Thermomicrobium roseum L-alanine dehydrogenase to synthesize L-alanine derivatives, Enzyme and Microbial Technology, 2023, 169, 110265. "NADP+ dependent formate dehydrogenase: A Review" Biocatalysis and Biotransformation, 2020, 39(4), 260-268.

Response: Dehydrogenases are a huge group of enzymes, each individual representative of this group, for example, glutamate dehydrogenase, alanine dehydrogenase, formate dehydrogenase, etc., requires a comprehensive description and, in our opinion, a summary table could overload the Introduction section and shift the emphasis to a topic that is not directly related to the topic of the manuscript. Nevertheless, we are very grateful to the reviewer for the idea with the table, and added the table dedicated to the comparison of GDHs with different cofactor specificities in the Results and discussion section (new Table 3, p. 18). Moreover, as recommended, we discussed importance of NAD(P) dependence for dehydrogenases. We have placed this information in the Conclusion section pp. 26-27, lines 624-630.

Comment:-The selection of buffers (MES, imidazole, Tris-HCl, CHES) is appropriate, but the rationale for using pH 6.0 for NADP+ and pH 9.5 for NAD+ in oxidative deamination should be explicitly discussed.

Response: pH conditions for GDH activity assay were selected based on pH optimum. The respective sentence was added in the manuscript, p. 16, lines 370-371. Moreover, in the original version of S2 Fig we found an error in the X scale values and corrected it (as we understand it, it was this inaccuracy in the graph that caused the reviewer's concern), we also added a statistical analysis of the results to the graph (new S2 Fig). Also, we added discussion about possible reason of different pH optima for NAD and NADP-dependent enzymatic reactions, p. 20, lines 451-467. We suggest that His250 of GdhPa may act as a pH-dependent “switch” for redox cofactor specificity, and this mechanism may underlie the dual coenzyme specificity and explain the different pH optima for NAD+- and NADP+-dependent reactions (9.5 and 6.0–6.5, respectively).

Comment:-It would be helpful to compare kinetic parameters to a broader set of bacterial GDHs, emphasizing how GdhPa's properties differ functionally.

Response: Kinetic parameters of different bacterial GDHs and GdhPa were compared, pp. 17-18, lines 395-414; new Table 3.

Comment:- The mechanistic insights into dual coenzyme specificity could be explored further, possibly through structural analysis or computational modeling. tructural modeling of GdhPa using Alphafold2 or SWISS-MODEL would provide deeper insights into coenzyme binding and active site interactions. Here a reference for reviewing, "Structural insights into the NAD+-dependent formate dehydrogenase mechanism revealed from the NADH complex and the formate NAD+ ternary complex of the Chaetomium thermophilum enzyme" Structural Biology, 2020, 212(3),107657.

Response: Thank you for this idea! We performed 3D structural modeling and discussed dual coenzyme specificity of investigated enzyme based on the obtained models, pp. 18-20, lines 415-467; new Fig 5.

Comment:-The statistical significance of differences should be clearly indicated (p-values, error bars clarification, etc.).

Response: A detailed statistical analysis of results was conducted, significance parameters were determined for comparisons against the control group as shown in new figures 6, 7, 8, S2, S3 and their legends (lines 521-525; 542-545; 599-600; 880-881; 886-889), and in p. 22, lines 514-517.

Reviewer #2:

Response: The authors would like to express their gratitude to Reviewer #2 for the valuable comments and constructive suggestions provided for the revision of the manuscript. We have carefully addressed all remarks, and our detailed responses are presented below

Content-related revisions made in response to the reviewers' comments are highlighted in yellow in the file titled "Revised Manuscript_partially_marked." Additionally, all corrections—including both content-related changes and minor grammatical/stylistic edits—are tracked in the "Revised Manuscript with Track Changes" file, generated using Microsoft Word’s Compare Documents feature.

The line numbers mentioned in the response correspond to the documents "Manuscript” and "Revised Manuscript_partially_marked".

Comment:An English language revision is required.

Response: The text has been thoroughly revised for English language accuracy using editing service LetPub (http://www.letpub.com). A large number of grammatical and stylistic corrections have been made as indicated in the file “Revised Manuscript with Track Changes”.

Comment:The Introduction, as well as the literature, require improvement: mention of plant GDHs and their evolution must be included (see for reference: Kwinta, J., Bielawski, W. Glutamate dehydrogenase in higher plants. Acta Physiol Plant 20, 453–463 (1998). https://doi.org/10.1007/s11738-998-0033-1. Tercé-Laforgue et al. Resolving the role of plant glutamate dehydrogenase: II. Physiological characterization of plants overexpressing the two enzyme subunits individually or simultaneously. Plant Cell Physiol. 2013 Oct;54(10):1635-47. doi: 10.1093/pcp/pct108. Brambilla et al. Glutamate dehydrogenase in “Liverworld”—A study in selected species to explore a key enzyme of plant primary metabolism in Marchantiophyta. Physiologia Plantarum 2023, 175, doi: 10.1111/ppl.14071)

Response: As requested, we added information about plant GDHs pp. 3-4, lines 54-56; 63-71

Results:

Comment: 1) statistical analysis and significance must be performed -and provided - for the enzyme activity results

Response: A detailed statistical analysis of results was conducted, significance parameters were determined for comparisons against the control group as shown in new figures 6, 7, 8, S2, S3 and their legends (lines 521-525; 542-545; 599-600; 880-881; 886-889), and in p. 22, lines 514-517.

Comment: 2) a more detailed discussion about the conserved aminoacids in functionally critical positions must be provided

Response: We performed 3D structural modeling and discussed dual coenzyme specificity of investigated enzyme based on the obtained models; conserved amino acids in functionally critical positions and their role in coenzyme specificity were discussed, pp. 18-20, lines 415-467; new Fig 5.

Comment: 3) in 2025, no newly characterized GDHs can lack an in gel activity characterization, as zymogram is the only way to reveal and assess the distribution of GDH isoenzymes in their active form. This is my major point, and it's not questionable....

Response: We would like to clarify that, unlike plant cells, the studied bacterial strain contains only a single gene encoding GDH, making the presence of isoenzymes unlikely. The requested experiment was done, which confirmed the GDH activity and dual coenzyme specificity of the studied enzyme, pp. 15-16, lines 342-362; new Fig 4. The performed zymogram analysis allowed us to determine the size of the active form of the protein, while mass spectrometry confirmed that the corresponding protein bands excised from the gel contained solely GdhAPa.

Comment: Please fix "Mechaelis constants" with "Mechaelis-Menten constant"

Response: Done, "Michaelis" was replaced with "Michaelis-Menten" throughout the text.

Comment: Figure 4 and 5: please delete "additives"

Response: Done in new Fig 6 and Fig 7.

Comment: Figure S1: please provide a less slovenly image

Response: Done in new S1 Fig.

Comment: Figure S2: data are presented without standard deviation bars and statistically significance marks. Please provide a more scientific graph.

Response: Done in new S2 Fig.

Comment: Figure S3: standard deviation bars curiously seem too homogenous among each others.... My feeling is that bars do not represent the real deviation from means, as the statistical analysis is apparently lacking. Please provide the relative test applied and the significance

Response: Done. The Brown-Forsythe and Welch ANOVA followed by Dunnett's post-hoc test were applied to determine statistically significant difference as indicated in new S3 Fig and other new figures 6, 7, 8, S2, and their legends (lines 521-525; 542-545; 599-600; 880-881; 886-889), and in p. 22, lines 514-517.

Comment: Figure S5: in my opinion. it should be embedded in the main text

Response: Thank you for this idea. We added this figure as new Fig 8 in the main text.

Comment: Supporting Table S1: primer's names are quite weird, as normally they are not marked with + o - symbols rather than with F o Fw (for forward) and R o Rev (for Reverse)

Response: We changed the primer names as requested, although we did not find any special rules regulating the names of primers.

---

## [Decision Letter · Decision Letter 1]

Dear Dr. Zakataeva,

Thank you for submitting your manuscript to PLOS ONE. After careful consideration, we feel that it has merit but does not fully meet PLOS ONE’s publication criteria as it currently stands. Therefore, we invite you to submit a revised version of the manuscript that addresses the points raised during the review process.

Certain critical details, as indicated by the reviewers, must be addressed prior to the acceptance of the manuscript.

We look forward to receiving your revised manuscript.

Kind regards,

Marcos Pileggi, Ph.D

Academic Editor

PLOS ONE

Journal Requirements:

Reviewers' comments:

Reviewer's Responses to Questions

**Comments to the Author**

Reviewer #1: (No Response)

Reviewer #2: (No Response)

2. Is the manuscript technically sound, and do the data support the conclusions?

Reviewer #1: (No Response)

Reviewer #2: Yes

3. Has the statistical analysis been performed appropriately and rigorously?

Reviewer #1: (No Response)

Reviewer #2: Yes

4. Have the authors made all data underlying the findings in their manuscript fully available?

Reviewer #1: (No Response)

Reviewer #2: No

5. Is the manuscript presented in an intelligible fashion and written in standard English?

Reviewer #1: (No Response)

Reviewer #2: Yes

Reviewer #1: (No Response)

Reviewer #2: Overall, the Authors responded to the comments, and the manuscript has improved significantly as a result.

However, they missed to address one of my major concerns.

In fact, given the complexity of the functional role of GDH enzyme, which is still to be clarified, it is necessary that the authors provide at least a glimpse into its evolutionary features. That's why I suggested to improve the literature by mentioning thre of the most relevant and complete publications (Kwinta, J., Bielawski, W. Glutamate dehydrogenase in higher plants. Acta Physiol Plant 20, 453–463 (1998). https://doi.org/10.1007/s11738-998-0033-1. Tercé-Laforgue et al. Resolving the role of plant glutamate dehydrogenase: II. Physiological characterization of plants overexpressing the two enzyme subunits individually or simultaneously. Plant Cell Physiol. 2013 Oct;54(10):1635-47. doi: 10.1093/pcp/pct108. Brambilla et al. Glutamate dehydrogenase in “Liverworld”—A study in selected species to explore a key enzyme of plant primary metabolism in Marchantiophyta. Physiologia Plantarum 2023, 175, doi: 10.1111/ppl.14071). Despite that, Authors only add one of the suggestions, completely ignoring the sense of the comment itself. Thus, the introduction and the discussion about the conserved aminoacids in functionally critical positions are not adequately informative.

**Do you want your identity to be public for this peer review?** For information about this choice, including consent withdrawal, please see our Privacy Policy

Reviewer #1: No

Reviewer #2: No

---

## [Author Response · Author response to Decision Letter 2]

25 Jun 2025

We appreciate the time and effort invested by the academic editor and reviewers in evaluating our manuscript.

Journal Requirements:

Answer. We have carefully reviewed the reference list and in-text citations to ensure compliance with the journal’s guidelines. After some corrections in the text and thorough verification, we confirm that the reference list does not contain any retracted articles.

In response to the reviewer’ comments, the following additional corrections have been made:

1. Additional References: We have incorporated several new references to strengthen the manuscript.

2. Updated Numbering: Due to these additions, the in-text citation numbers and the reference list order have been adjusted accordingly.

All modifications, including changes to citations and the reference list, are clearly highlighted in the tracked-changes version of the manuscript ("Revised Manuscript with Track Changes").

Reviewers' comments:

Reviewer #1: (No Response)

Reviewer #2: Overall, the Authors responded to the comments, and the manuscript has improved significantly as a result.

However, they missed to address one of my major concerns.

In fact, given the complexity of the functional role of GDH enzyme, which is still to be clarified, it is necessary that the authors provide at least a glimpse into its evolutionary features. That's why I suggested to improve the literature by mentioning thre of the most relevant and complete publications (Kwinta, J., Bielawski, W. Glutamate dehydrogenase in higher plants. Acta Physiol Plant 20, 453–463 (1998). https://doi.org/10.1007/s11738-998-0033-1. Tercé-Laforgue et al. Resolving the role of plant glutamate dehydrogenase: II. Physiological characterization of plants overexpressing the two enzyme subunits individually or simultaneously. Plant Cell Physiol. 2013 Oct;54(10):1635-47. doi: 10.1093/pcp/pct108. Brambilla et al. Glutamate dehydrogenase in “Liverworld”—A study in selected species to explore a key enzyme of plant primary metabolism in Marchantiophyta. Physiologia Plantarum 2023, 175, doi: 10.1111/ppl.14071). Despite that, Authors only add one of the suggestions, completely ignoring the sense of the comment itself. Thus, the introduction and the discussion about the conserved aminoacids in functionally critical positions are not adequately informative.

Answer. We thank Reviewer #2 for their careful reading of our manuscript and for valuable recommendations to improve it. We apologize for interpreting the initial comment (“The Introduction, as well as the literature, require improvement: mention of plant GDHs and their evolution must be included”) too literally.

We have now addressed the reviewer’s request by the following:

1. Citing all three recommended articles.

2. Expanding the discussion on the evolution of GDH enzymes in different species (lines 55–60, 67–86; 496–498).

3. Adding a new S1 Fig, which presents a phylogenetic tree of glutamate dehydrogenases from various organisms, including plants, animals, bacteria, and Archaea. We also discuss the phylogenetic clustering of P. ananatis GDHs within a NAD/NADP-dependent clade distinct from NADP-specific GDHs, possibly indicating horizontal gene transfer (lines 344–359, 902–911).

We believe the revised manuscript now addresses the reviewers' concerns while maintaining clarity and scientific rigor. Please let us know if further modifications are required. Thank you for your time and consideration.

---

## [Decision Letter · Decision Letter 2]

Glutamate dehydrogenase from Pantoea ananatis: A new bacterial enzyme with dual coenzyme specificity

PONE-D-25-07402R2

Dear Dr. Zakataeva,

We’re pleased to inform you that your manuscript has been judged scientifically suitable for publication and will be formally accepted for publication once it meets all outstanding technical requirements.

Kind regards,

Marcos Pileggi, Ph.D

Academic Editor

PLOS ONE

Additional Editor Comments (optional):

Reviewers' comments:

Reviewer's Responses to Questions

**Comments to the Author**

Reviewer #2: All comments have been addressed

2. Is the manuscript technically sound, and do the data support the conclusions?

Reviewer #2: Yes

3. Has the statistical analysis been performed appropriately and rigorously?

Reviewer #2: Yes

4. Have the authors made all data underlying the findings in their manuscript fully available?

Reviewer #2: Yes

5. Is the manuscript presented in an intelligible fashion and written in standard English?

Reviewer #2: Yes

Reviewer #2: (No Response)

**Do you want your identity to be public for this peer review?** For information about this choice, including consent withdrawal, please see our Privacy Policy

Reviewer #2: No
